# Few-Shot Learning from Gigapixel Images via Hierarchical Vision-Language Alignment and Modeling

**Bryan Wong[1]*  Jong Woo Kim[1]*  Huazhu Fu[2]  Mun Yong Yi[1]†**
[1]KAIST    [2]IHPC, A*STAR
{bryan.wong, gsds4885, munyi}@kaist.ac.kr
hzfu@ieee.org

## Abstract

Vision-language models (VLMs) have recently been integrated into multiple instance learning (MIL) frameworks to address the challenge of few-shot, weakly supervised classification of whole slide images (WSIs). A key trend involves leveraging multi-scale information to better represent hierarchical tissue structures. However, existing methods often face two key limitations: (1) insufficient modeling of interactions within the same modalities across scales (e.g., $5\times$ and $20\times$) and (2) inadequate alignment between visual and textual modalities on the same scale. To address these gaps, we propose **HiVE-MIL**, a hierarchical vision-language framework that constructs a unified graph consisting of (1) parent–child links between coarse ($5\times$) and fine ($20\times$) visual/textual nodes to **capture hierarchical relationships**, and (2) **heterogeneous intra-scale edges** linking visual and textual nodes on the same scale. To further enhance semantic consistency, HiVE-MIL incorporates a two-stage, text-guided dynamic filtering mechanism that removes weakly correlated patch–text pairs, and introduces a hierarchical contrastive loss to align textual semantics across scales. Extensive experiments on TCGA breast, lung, and kidney cancer datasets demonstrate that HiVE-MIL consistently outperforms both traditional MIL and recent VLM-based MIL approaches, achieving gains of up to 4.1% in macro F1 under 16-shot settings. Our results demonstrate the value of jointly modeling hierarchical structure and multimodal alignment for efficient and scalable learning from limited pathology data. The code is available at `https://github.com/bryanwong17/HiVE-MIL`.

## 1  Introduction

Whole slide image (WSI) classification is a central task in computational pathology (CPath), enabling cancer diagnosis, subtyping, and prognosis prediction [50, 51, 10]. With gigapixel resolution, WSIs contain detailed spatial information ranging from coarse tissue structures to fine-grained cellular morphology [20], which is crucial for accurate interpretation in these diagnostic tasks. To handle their large size and the absence of fine-grained annotations, multiple instance learning (MIL) is widely adopted [8, 28]. In MIL, each WSI is treated as a bag of instances (patches), and a slide-level label is predicted through a feature aggregator with supervision only at the slide level (Figure 1(A)). However, traditional MIL models [27, 33, 38, 47, 54, 34] face key challenges in real-world clinical settings [23]: (1) they rely on large labeled datasets, which are difficult to obtain due to privacy concerns [40] and the rarity of certain diseases [32], making them ineffective in few-shot scenarios where labeled

---

*Equal contribution.

†Corresponding author.

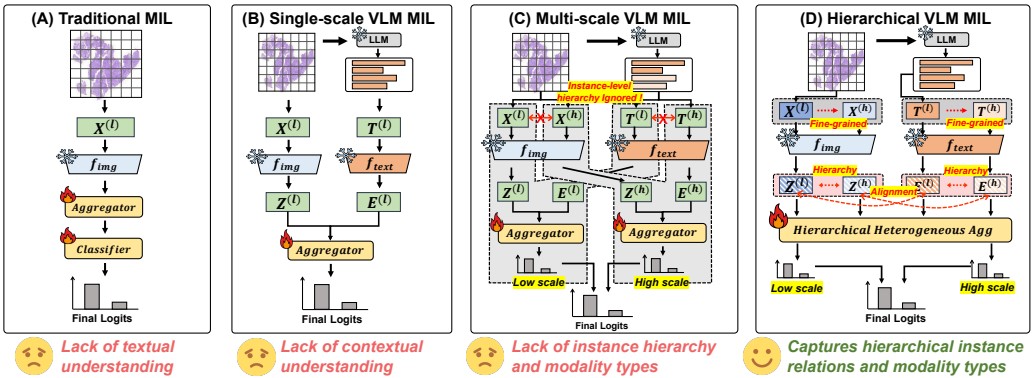

Figure 1: Comparison of four paradigms for FSWC. (A) Traditional MIL uses only visual features and requires extensive WSI labels. (B) Single-scale VLM-MIL introduces LLM-based prompts but lacks contextual and scale-aware reasoning. (C) Multi-scale VLM-MIL adds scale-specific prompts but lacks structured modeling of hierarchical dependencies and modality-aware interactions (green indicates missing modality-specific modeling). (D) HiVE-MIL **(Ours)** explicitly captures coarse-to-fine hierarchical relationships and aligns visual and textual features at each scale, enabling efficient and scalable learning.

WSIs are scarce [44, 48]; (2) they use only visual features, making them highly sensitive to staining variability [36, 11] and domain shifts [37, 48]. These challenges call for a **data-efficient approach** that leverages prior domain knowledge for robust learning under limited supervision.

Vision-language models (VLMs) such as CLIP [46], BLIP [35], and Flamingo [2] have demonstrated strong zero- and few-shot transfer performance by learning joint image-text embeddings through contrastive learning [30, 55, 49, 12]. However, their training in natural images and generic captions limits their effectiveness in CPath, where domain-specific semantics are critical. To address this issue, domain-adapted VLMs such as PLIP [25], QuiltNet [26], and CONCH [39] leverage large-scale pathology patch-text datasets and improve robustness and generalization at the patch level. Motivated by these capabilities, recent efforts [44, 45, 19] have integrated them into MIL frameworks to address the few-shot weakly-supervised WSI classification (FSWC) problem by incorporating domain knowledge via text prompts (Figure 1(B)). Multi-scale VLM-based MIL methods [48, 22, 41] further enhance single-scale models by using scale-specific prompts that capture WSI multi-scale information to better represent hierarchical tissue structures.

Although recent multiscale VLM-based MIL methods [48, 22, 41] have made impressive progress, effectively transferring VLM knowledge to MIL remains a serious challenge due to limited modeling of the complex hierarchical structure of WSIs and insufficient integration of multiple modalities. Specifically, as illustrated in Figure 1(C), existing methods face two key limitations. **(1) Insufficient modeling of hierarchical interactions within the same modalities.** Existing models process visual and textual features independently at each scale and combine them through simple summation or averaging at the final prediction stage. This naïve fusion fails to capture hierarchical relationships across scales within each modality. In the visual domain, it overlooks the semantic progression from coarse tissue-level patterns to fine-grained cellular morphology. Also, in the textual domain, it fails to represent the transition from general morphological descriptions to specific structural details, limiting the model's ability to leverage hierarchical semantics effectively. **(2) Inadequate alignment between modalities on the same scale.** Existing models do not *fully* explore interactions between modalities when constructing task-specific knowledge, often relying on simpler alternative mechanisms that lack the strong inductive bias [4, 58] needed for fine-grained cross-modal alignment. This limits their ability to effectively integrate visual and textual features.

To this end, we propose **HiVE-MIL** (**Hi**erarchical **V**ision-Languag**E MIL**), a unified framework that explicitly models hierarchical relationships within modalities and intra-scale alignments across modalities in multi-scale vision-language settings (Figure 1(D)). **(1) To capture the hierarchical interactions across scales**, HiVE-MIL constructs *hierarchical edges* between visual nodes and between textual nodes across coarse ($5\times$) and fine ($20\times$) scales based on parent–child relationships,

and jointly introduces a *Modality-Scale Attention (MSA)* mechanism that handles these connections, allowing the model to represent semantic progression from global context to localized detail while preserving hierarchical consistency. To ensure semantic coherence in textual space, HiVE-MIL incorporates a *Hierarchical Text Contrastive Loss (HTCL)* that aligns class-level text embeddings across scales. Unlike prior methods that fuse multi-scale features only at the output, HiVE-MIL facilitates explicit hierarchical interaction for more coherent representations. **(2) To effectively model intra-scale interactions across modalities**, HiVE-MIL utilizes a *heterogeneous graph* that captures semantic connections between visual and textual nodes on the same scale. This design improves the alignment quality between modalities and contributes to semantically coherent multimodal integration. Furthermore, to improve alignment accuracy, HiVE-MIL introduces a *Text-Guided Dynamic Filtering (TGDF)* module that filters out semantically irrelevant or weakly matched patch–text pairs, such as when a normal patch is mistakenly paired with IDC or ILC-related text, using text-wise soft thresholding. Together, these components enable HiVE-MIL to model hierarchical and semantic dependencies across scales and modalities, improving robustness and accuracy in FSWC.

**The main contributions of this work are as follows:**

• We construct a hierarchical graph with hierarchical edges between coarse ($5\times$) and fine ($20\times$) visual/textual nodes via parent–child links, and introduce *Modality-Scale Attention (MSA)* and *Hierarchical Text Contrastive Loss (HTCL)* to enforce text semantic consistency across scales.
• We build a heterogeneous graph to connect visual and textual nodes on the same scale and apply a *Text-Guided Dynamic Filtering (TGDF)* module to remove weak or irrelevant patch–text pairs, improving intra-scale alignment.
• Extensive experiments on three real-world WSI datasets, including lung, breast, and kidney cancers, show that HiVE-MIL consistently outperforms traditional MIL and VLM-based MIL baselines across diverse few-shot settings and pathology foundation models.

## 2 Related Work

### 2.1 Multiple Instance Learning in CPath

WSI classification is typically formulated as a weakly supervised learning task in the MIL setting, where each slide is treated as a bag of unlabeled patches with supervision provided only at the slide level [14]. Embedding-based MIL approaches [27, 33, 38, 47, 54] are generally more effective than instance-based models [7, 9, 43], as they learn discriminative patch embeddings for aggregation [6]. Early methods rely on non-parametric aggregators (e.g., mean, max), while attention-based techniques such as ABMIL [27], DSMIL [33], and CLAM [38] introduce mechanisms to weigh patch relevance. TransMIL [47] captures spatial dependencies via self-attention, while DTFD-MIL [54] employs a double-tier distillation framework with pseudo-bag supervision. Graph-based models [57, 17, 24, 34] improve contextual modeling by constructing structured graphs that capture spatial and relational dependencies between instances. Existing methods remain ineffective for FSWC due to their reliance on large labeled datasets and visual-only features, which make them sensitive to staining variability [36, 11] and domain shifts [37, 48], highlighting the need for a data-efficient approach that leverages prior domain knowledge for robust learning under limited supervision.

### 2.2 Vision-Language Models in CPath

Vision-language foundation models such as CLIP [46], BLIP [35], and Flamingo [2] learn joint image-text embeddings through contrastive learning and enable zero- and few-shot transfer via prompting [30, 55, 49, 12]. In CPath, PLIP [25], QuiltNet [26], and CONCH [39] adapt these models using large-scale pathology image-text datasets to enhance robustness in patch-level tasks. Motivated by their few-shot capabilities, recent works extend VLMs to MIL settings for WSI classification. However, adaptation methods developed for natural images, such as CoOp [59], CLIP-Adapter [15], and HeGraphAdapter [56], overlook the gigapixel scale, hierarchical structure, and fine-grained semantics of WSIs, limiting their effectiveness in weakly supervised scenarios. To address this issue, several studies integrate VLMs into MIL frameworks [44, 45, 48, 22, 19]. TOP [44] and FOCUS [19] adopt prompt-based supervision and multi-stage compression, respectively, but are limited to single-scale inputs and cannot capture multi-scale context. Multi-scale VLM-based MIL methods [48, 22, 41] improve upon these approaches by introducing scale-specific prompts that reflect the hierarchical nature of WSIs, enabling more context-aware and semantically aligned representations.

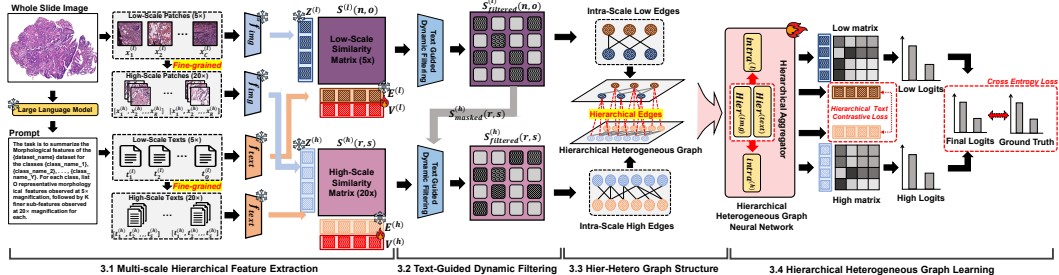

Figure 2: Overview of the proposed HiVE-MIL framework.

Nevertheless, they still face two key challenges: they fail to model semantic progression across scales within each modality and they lack alignment between visual and textual features on the same scale, weakening semantic grounding and multimodal integration. Unlike previous work, HiVE-MIL explicitly addresses both limitations by modeling hierarchical relationships across scales and aligning visual and textual features within each scale, enabling more robust performance in FSWC.

## 3 Method

### 3.1 Multi-scale Hierarchical Feature Extraction

Unlike prior work [48, 22, 41] that employs multi-scale text prompts and patch features without modeling explicit hierarchical relationships, our method organizes visual and textual representations into a *parent–child* hierarchy, where each fine-scale (child) node is explicitly linked to its corresponding coarse-scale (parent) node. This hierarchical design reflects the diagnostic workflow of pathologists and captures the intrinsic multi-scale structure of WSIs.

**Hierarchical Visual Feature Extraction.** Each WSI is divided into non-overlapping patches at two scales: low-scale ($5\times$) and high-scale ($20\times$). We extract $N$ low-scale patches $x_n^{(l)}$ and encode them using a frozen VLM image encoder, resulting in visual features $z_n^{(l)} = f_{\text{img}}(x_n^{(l)}) \in \mathbb{R}^D$. To capture finer-grained details within each low-scale patch, we subdivide it into $M = (20/5)^2 = 16$ high-scale patches $x_{n,m}^{(h)}$, arranged in a $4 \times 4$ spatial grid. Here, $m \in \{1, \ldots, M\}$ denotes the relative position of each high-scale patch within the grid. If fewer than 16 high-scale patches are available (e.g., due to whitespace), zero embeddings are inserted to preserve consistent feature dimensions. Each valid high-scale patch is encoded as $z_{n,m}^{(h)} = f_{\text{img}}(x_{n,m}^{(h)}) \in \mathbb{R}^D$. To simplify notation, we flatten the hierarchical indices as $r = (n, m)$, resulting in the final encoded representation $z_r^{(h)}$. This yields hierarchical visual features: $Z^{(l)} \in \mathbb{R}^{N \times D}$ on the low scale and $Z^{(h)} \in \mathbb{R}^{R \times D}$ on the high scale. These representations maintain spatial alignment across scales and serve as the basis for hierarchical modeling. Please refer to the Appendix A.2 for hierarchical patch extraction details.

**Hierarchical Textual Feature Extraction.** Tumor heterogeneity causes WSIs within the same class to exhibit spatially diverse morphological features, including coarse tissue-level structures (e.g., *Glandular Acinar Patterns*) and fine-grained cellular traits (e.g., *Nuclear Hyperchromasia*), forming a hierarchical pattern. To capture this, we use an LLM to generate hierarchical textual prompts based on the following template:

---

**LLM Prompt Template**

The task is to summarize the morphological features of the {dataset_name} dataset for the {class_name_1},...,{class_name_c} classes. For each class, list **O** representative morphological features observed at **5**$\times$ magnification, followed by **K** finer sub-features observed at **20**$\times$ magnification for each. Each description should include the morphological term along with an explanation of its defining visual features.

---

This yields hierarchical textual descriptions: *Low-scale Text$_o$* and *High-scale Text$_{o,k}$*, where each high-scale (child) text is associated with its corresponding low-scale (parent) text, enabling semanti-

cally consistent hierarchical alignment. Following CoOp [59], each text is prepended with $L$ learnable tokens $v^{(*)} \in \mathbb{R}^{L \times D}$, resulting in prompt embeddings:

$$t_o^{(l)} = [v_1^{(l)}] \dots [v_L^{(l)}][\textit{Low-scale Text}_o] \quad t_{o,k}^{(h)} = [v_1^{(h)}] \dots [v_L^{(h)}][\textit{High-scale Text}_{o,k}] \tag{1}$$

These are encoded via a frozen VLM text encoder as $e_o^{(l)} = f_{\text{text}}(t_o^{(l)})$ and $e_{o,k}^{(h)} = f_{\text{text}}(t_{o,k}^{(h)})$. For notational simplicity, we flatten the indices to $s = (o, k)$, producing hierarchical textual embeddings $E^{(l)} \in \mathbb{R}^{O \times D}$ and $E^{(h)} \in \mathbb{R}^{S \times D}$. For examples of hierarchical text generated by the LLM, please refer to the Appendix A.3.

## 3.2 Text-Guided Dynamic Filtering

We propose a two-stage *Text-Guided Dynamic Filtering (TGDF)* module that improves intra-scale visual–textual alignment by removing semantically irrelevant image–text pairs in a *top-down* manner. In the first stage, TGDF filters out low-scale patches (e.g., normal tissue) and patch–text pairs with low similarity to the low-scale text. In the second stage, it refines high-scale patch selection based on retained low-scale patches and filters out weakly aligned high-scale patch–text pairs. This prevents irrelevant patches from being connected to disease-specific prompts, which could confuse the model. The remaining meaningful visual–textual pairs are then used to guide intra-scale edge construction in the heterogeneous graph. See Appendix D for the detailed algorithm and pseudocode.

**Stage 1 (Low-scale Filtering).** We compute the cosine similarity between low-scale patch features $Z^{(l)} \in \mathbb{R}^{N \times D}$ and low-scale text features $E^{(l)} \in \mathbb{R}^{O \times D}$, forming a similarity matrix $S^{(l)} \in \mathbb{R}^{N \times O}$. To discard weak or irrelevant matches, we apply text-wise soft thresholding by computing the mean $\mu_o$ and standard deviation $\sigma_o$ across all patches in the WSI for each text $o$, where $\alpha$ controls the filtering sensitivity (Eq. 2 and 3).

$$S_{\text{filtered}}^{(l)}(n, o) = \mathbb{I}\left(S^{(l)}(n, o) \geq \mu_o + \alpha \cdot \sigma_o\right) \tag{2}$$

The filtered similarity matrix $S_{\text{filtered}}^{(l)} \in \mathbb{R}^{N \times O}$ implicitly serves as a soft relevance mask, where *non-zero* entries indicate semantically valid patch-text pairs. This matrix is then propagated to guide high-scale filtering.

**Stage 2 (High-scale Refinement).** For each retained low-scale patch $n$ and associated text $o$, we use the corresponding high-scale image patches $z_r^{(h)}$ and submorphology texts $e_s^{(h)}$, and compute a similarity matrix $S^{(h)} \in \mathbb{R}^{R \times S}$. To maintain consistency with the first stage filtering, we mask irrelevant pairs using the filtered similarity score: $S_{\text{masked}}^{(h)}(r, s) = S^{(h)}(r, s) \cdot S_{\text{filtered}}^{(l)}(n, o)$, where $r = (n, m)$ and $s = (o, k)$. We then apply text-wise soft thresholding to $S_{\text{masked}}^{(h)}$ by computing the mean $\mu_s$ and standard deviation $\sigma_s$ for each submorphology text $s$.

$$S_{\text{filtered}}^{(h)}(r, s) = \mathbb{I}\left(S_{\text{masked}}^{(h)}(r, s) \geq \mu_s + \alpha \cdot \sigma_s\right) \tag{3}$$

The final filtered similarity matrices at both low and high scales identify semantically aligned patch-text pairs at each scale. These aligned pairs are then used to construct intra-scale edges between modalities at both scales in the graph. As text tokens are updated during training (Eq. 1), the resulting text features and similarity matrices vary, enabling dynamic filtering and non-fixed intra-scale edges.

## 3.3 Hierarchical Heterogeneous Graph Structure

To enable integration across multiple scales and modalities, we propose a *Hierarchical Heterogeneous Graph (HHG)*, $\mathcal{G}_{\mathcal{HHG}} = (\mathcal{V}, \mathcal{E}, \mathcal{T}, \mathcal{R})$. This graph encodes intra-scale semantics and hierarchical structure in a modality-aware manner, allowing each node to play different roles depending on its modality. The edge set $\mathcal{E} = \mathcal{E}^{\text{intra}} \cup \mathcal{E}^{\text{hier}}$ supports structured message passing across both scales and modalities. Detailed definitions of the node set $\mathcal{V}$ and the edge set $\mathcal{E}$ are provided below.

**Node Set.** We define node types as $\mathcal{T} = \{\text{img}^{(l)}, \text{img}^{(h)}, \text{text}^{(l)}, \text{text}^{(h)}\}$, where each type specifies both the modality (image or text) and the scale (low or high). The full node set is:

$$\mathcal{V} = \{z_n^{(l)}\}_{n=1}^N \cup \{z_r^{(h)}\}_{r=1}^R \cup \{e_o^{(l)}\}_{o=1}^O \cup \{e_s^{(h)}\}_{s=1}^S \tag{4}$$

**Edge Set.** We define relation types as $\mathcal{R} = \{\text{intra}^{(l)}, \text{intra}^{(h)}, \text{hier}^{(\text{img})}, \text{hier}^{(\text{text})}\}$. Intra-scale edges connect *valid* patch and text nodes on the same scale using the TGDF-filtered similarity matrix (Section 3.2). Hierarchical edges capture hierarchical alignment within each modality: $\text{hier}^{(\text{img})}$ links low-scale patch nodes $z_n^{(l)}$ to high-scale ones $z_r^{(h)}$ via absolute coordinate mapping; $\text{hier}^{(\text{text})}$ is constructed analogously using hierarchical text structure (Section 3.1). All edges are bidirectional $(a \leftrightarrow b)$ to support relation-aware message passing across scales and modalities. The edge set is defined as:

$$
\mathcal{E}^{\text{intra}} = \underbrace{\{z_n^{(l)} \leftrightarrow e_o^{(l)} \mid S_{\text{filtered}}^{(l)}(n, o) > 0\}}_{\text{intra}^{(l)}} \cup \underbrace{\{z_r^{(h)} \leftrightarrow e_s^{(h)} \mid S_{\text{filtered}}^{(h)}(r, s) > 0\}}_{\text{intra}^{(h)}},
$$
$$
\mathcal{E}^{\text{hier}} = \underbrace{\{z_n^{(l)} \leftrightarrow z_r^{(h)}\}}_{\text{hier}^{(\text{img})}} \cup \underbrace{\{e_o^{(l)} \leftrightarrow e_s^{(h)}\}}_{\text{hier}^{(\text{text})}}
\tag{5}
$$

## 3.4 Hierarchical Heterogeneous Graph Learning

We introduce a hierarchical heterogeneous graph neural network (HHGNN) designed to operate on the constructed HHG. HHGNN performs relation-specific message passing to model local semantic interactions within each scale and propagate hierarchical signals across scales, enabling robust representation learning on multi-modal, multi-scale graphs. Details of the message passing are provided in Appendix E.

**Intra-scale Aggregator.** To capture intra-scale relationships between patch and text nodes on the same scale, we apply a relation-specific GraphSAGE [21] operator $\text{SAGE}^{(r)}(v)$ for each edge type $r \in \mathcal{R}^{\text{intra}}$ ($z_n^{(l)} \leftrightarrow e_o^{(l)}$, $z_r^{(h)} \leftrightarrow e_s^{(h)}$). The intra-scale representation is then computed by averaging the outputs over all such relations: $h_v^{\text{intra}} = \text{MEAN}(\{\text{SAGE}^{(r)}(v) \mid r \in \mathcal{R}^{\text{intra}}\})$. Initial node features $h_v$ are modality-specific embeddings.

**Hierarchical Aggregator.** To capture hierarchical interactions across both modalities and scales, we introduce *Modality-Scale Attention (MSA)*, an attention mechanism applied to hierarchical edges $r \in \mathcal{R}^{\text{hier}}$ ($z_n^{(l)} \leftrightarrow z_r^{(h)}$, $e_o^{(l)} \leftrightarrow e_s^{(h)}$). Each edge encodes both the scale direction and the modality type. The node features are first enhanced with scale embeddings and projected into query, key, and value vectors using relation-specific weights: $q_v = W_q^{(r)}(h_v + s_v)$, $k_u = W_k^{(r)}(h_u + s_u)$, and $v_u = W_v^{(r)}(h_u + s_u)$. The attention weights are computed as $\beta_{vu} = \text{softmax}\left(\frac{q_v^\top k_u}{\sqrt{d}}\right)$, and the final output is:

$$
h_v^{\text{hier}} = q_v + \sum_{u \in \mathcal{N}_r(v)} \beta_{vu} v_u
\tag{6}
$$

### 3.4.1 Feature Update and Classification

The final node representation is computed as $h_v = h_v^{\text{intra}} + h_v^{\text{hier}}$, combining intra-scale and hierarchical information. Let $\mathcal{V}_{\text{img}}^{(s)}$ and $\mathcal{V}_{\text{text}}^{(s)}$ denote the sets of patch and text nodes at scale $s \in \{l, h\}$, with feature matrices $\mathbf{X}^{(s)} = \{h_v \mid v \in \mathcal{V}_{\text{img}}^{(s)}\}$ and $\mathbf{T}^{(s)} = \{h_v \mid v \in \mathcal{V}_{\text{text}}^{(s)}\}$. Class-wise logits are computed by:

$$
\text{logit}_c^{(s)} = \frac{\gamma}{|I_c|} \sum_{i \in I_c} \text{TopKAvg}_{k^{(s)}}\left(\left[X^{(s)}\left(T^{(s)}\right)^\top\right]_{\cdot, i}\right)
\tag{7}
$$

where $I_c$ denote the class-specific text index set, where each $i \in I_c$ corresponds to a text prompt associated with class $c$, $\gamma$ the logit scaling factor provided by the VLM, and $k^{(s)}$ the number of top similarity scores considered at scale $s$. The operator $\text{TopKAvg}_{k^{(s)}}(\cdot)$ computes the average of top-$k^{(s)}$ scores from the $i$-th text $(\cdot, i)$ in the scale-specific image–text similarity matrix, thereby aggregating signals from the most relevant images for each text prompt before the final summation.

### 3.4.2 Training Objectives

**Hierarchical Text Contrastive Loss (HTCL).** To encourage semantic alignment of morphological text embeddings across scales, we compute cosine similarity between low- and high-scale textual embeddings: $\text{sim}_{o,s} = \cos(\mathbf{T}_o^{(l)}, \mathbf{T}_s^{(h)})$, where $o$ is a parent (low-scale) and $s$ is its child (high-scale). For each anchor $s$, positive pairs are defined as $\mathcal{P}_s = \{o \mid \text{cls}(\text{Parent}(s)) = \text{cls}(o)\}$ and negatives as $\mathcal{N}_s = \{o \mid \text{cls}(\text{Parent}(s)) \neq \text{cls}(o)\}$. The loss is computed as:

$$\mathcal{L}_{\text{HTCL}} = \frac{1}{N} \sum_{i=1}^{N} \left( -\frac{1}{|\mathcal{P}_s|} \sum_{j \in \mathcal{P}_s} \log \sigma(\text{sim}_{o,s}) - \frac{1}{|\mathcal{N}_s|} \sum_{j \in \mathcal{N}_s} \log \sigma(-\text{sim}_{o,s}) \right) \qquad (8)$$

Let $\mathbf{z}_i \in \mathbb{R}^C$ denote the final class-wise logits for sample $i$, obtained by summing the low- and high-scale outputs, i.e., $z_{i,c} = \text{logit}_{i,c}^{(l)} + \text{logit}_{i,c}^{(h)}$. The total loss is then:

$$\mathcal{L}_{\text{total}} = \mathcal{L}_{\text{CE}}(z_i, y_i) + \lambda \, \mathcal{L}_{\text{HTCL}} \qquad (9)$$

where $\mathcal{L}_{\text{CE}}$ is the standard cross-entropy loss, $y_i \in \{1, \ldots, C\}$ is the ground-truth WSI class label, and $\lambda$ balances the two loss terms.

## 4 Experiments

### 4.1 Experimental Settings

**Datasets.** We utilize three publicly available WSI datasets: TCGA-NSCLC (lung), TCGA-BRCA (breast), and TCGA-RCC (kidney), obtained from The Cancer Genome Atlas (TCGA).[3] Following [48], each dataset is split into training, validation, and test sets using a fixed 4:3:3 ratio. For the few-shot setting, we randomly sample 4, 8, and 16 WSIs per class from the training set. For detailed dataset statistics and preprocessing steps, please refer to the Appendix A.1.

**VLM and Baselines.** We evaluate using three pathology vision-language foundation models: PLIP [25], QuiltNet [26], and the recent CONCH [39]. Baselines include: (1) Pooling-based (max, mean); (2) Traditional MIL-based (ABMIL [27], DSMIL [33], CLAM-SB/MB [38], TransMIL [47], DTFD-MIL (AFS) [54], WiKG [34]); (3) Single-scale VLM-based MIL method (FOCUS [19]); (4) Multi-scale VLM-based MIL methods (ViLa-MIL [48], MSCPT [22]). All single-scale models including the pooling-based use $20\times$ patches, whereas multi-scale models use $5\times$ and $20\times$. Please refer to the Appendix I for more details.

**Implementation Details.** HiVE-MIL operates on $5\times$ and $20\times$ patches, using GPT-4o [1] to generate $O = 4$ coarse-level texts and $K = 3$ fine-level substructures per class. We use $L = 16$ learnable context tokens (Eq. 1) and apply the TGDF threshold $\alpha = 0.5$ (Eqs. 2, 3). The HHG consists of two layers and a 2-head in MSA. HTCL is used with $\lambda = 0.5$ (Eq. 9). We train using Adam optimizer [31] (learning rate: 1e−4, weight decay: 1e−5), batch size 1, for up to 50 epochs with early stopping (patience 10). All experiments are run using PyTorch [42] on a workstation with two NVIDIA RTX A100 GPUs. Please refer to the Appendix H for additional implementation details.

**Evaluation Metrics.** We report accuracy (ACC), area under the curve (AUC), and macro F1 score. To mitigate dataset split variability in few-shot settings, all experiments are repeated five times, reporting the mean and standard deviation. To assess whether HiVE-MIL captures hierarchical text semantic alignment (i.e., parent–child hierarchy in text), we introduce *Hit Ratio* (Section 4.4).

### 4.2 Main Results

We adopt the 16-shot evaluation setting as our main experimental setup, following [48, 22, 41], and report results across three TCGA datasets (NSCLC, BRCA, RCC) and three vision-language pathology foundation models (PLIP, QuiltNet, CONCH). As shown in Table 1, HiVE-MIL consistently achieves the best performance across all settings, outperforming both traditional and VLM-based MIL methods, including their single-scale and multi-scale variants. Compared to the state-of-the-art

---
[3] https://portal.gdc.cancer.gov/

Table 1: **16-shot** results on three datasets using three pathology VLMs. The best and second-best results are highlighted in **bold** and underlined. HiVE-MIL outperforms all baselines in all settings.

| | Dataset | TCGA NSCLC | | | TCGA BRCA | | | TCGA RCC | | |
|---|---|---|---|---|---|---|---|---|---|---|
| | Model | ACC | AUC | Macro F1 | ACC | AUC | Macro F1 | ACC | AUC | Macro F1 |
| **PLIP [25]** 208K Pathology Image-Text Pairs | Max Pooling | 55.00±3.88 | 57.33±4.63 | 53.96±4.86 | 57.29±3.23 | 62.33±2.94 | 53.68±6.91 | 66.82±6.94 | 80.58±6.68 | 61.38±8.68 |
| | Mean Pooling | 61.73±5.65 | 65.29±7.55 | 61.15±6.16 | 65.25±4.40 | 70.83±3.93 | 64.04±4.42 | 79.62±3.51 | 92.09±1.92 | 76.67±3.49 |
| | ABMIL [27] | 70.64±2.98 | 78.44±3.63 | 70.37±3.09 | 65.83±5.33 | 72.87±7.88 | 65.29±5.78 | 80.00±3.71 | 93.01±1.53 | 77.95±3.43 |
| | DSMIL [33] | 72.63±3.88 | 79.88±4.60 | 72.48±3.96 | 71.38±3.20 | 77.55±1.62 | 71.04±3.40 | 86.74±1.23 | 96.44±0.63 | 84.63±1.51 |
| | CLAM-SB [38] | 75.96±2.60 | 83.79±3.21 | 75.94±2.61 | 71.75±3.57 | 80.00±2.59 | 71.49±3.60 | 85.98±1.51 | 96.22±0.48 | 83.35±1.54 |
| | CLAM-MB [38] | 73.46±3.15 | 82.13±3.41 | 73.42±3.13 | 72.50±2.92 | 78.39±2.95 | 72.20±2.87 | 86.97±1.03 | 96.53±0.78 | 84.92±1.03 |
| | TransMIL [47] | 73.21±3.02 | 81.44±2.75 | 72.98±2.95 | 72.08±3.32 | 79.47±3.71 | 71.94±3.34 | 87.05±1.52 | 96.51±0.56 | 84.96±1.32 |
| | DTFD-MIL [54] | 72.95±3.40 | 79.79±4.65 | 72.91±3.39 | 71.25±2.68 | 78.91±3.16 | 70.86±2.76 | 86.74±0.79 | 95.94±0.62 | 84.86±1.45 |
| | WiKG [34] | 67.89±3.66 | 75.54±4.05 | 67.51±3.62 | 67.71±2.19 | 74.92±4.16 | 67.15±2.42 | 83.07±0.89 | 94.34±0.76 | 80.32±1.40 |
| | ViLa-MIL [48] | 74.17±1.01 | 80.63±2.37 | 73.90±1.15 | 71.04±6.92 | 78.42±5.86 | 70.56±6.98 | 85.06±2.13 | 95.53±0.97 | 82.51±2.30 |
| | MSCPT [22] | 76.86±1.85 | 84.93±1.59 | 76.82±1.89 | 72.71±2.90 | 79.78±4.14 | 72.58±2.81 | 86.21±0.54 | 95.84±0.45 | 84.20±0.81 |
| | FOCUS [19] | 71.73±5.52 | 78.21±5.93 | 71.65±5.51 | 71.66±5.60 | 78.19±4.51 | 71.36±5.69 | 87.82±1.69 | 96.73±0.70 | 85.54±1.87 |
| | HiVE-MIL | **80.13±4.73** | **87.28±2.76** | **80.08±4.73** | **75.21±3.51** | **83.19±4.72** | **74.99±3.67** | **88.89±1.36** | **97.58±0.41** | **87.18±1.78** |
| | Δ from 2nd-best | (+3.27) | (+2.35) | (+3.26) | (+2.50) | (+3.19) | (+2.41) | (+1.07) | (+0.85) | (+1.64) |
| **QuiltNet [26]** 1M Pathology Image-Text Pairs | Max Pooling | 53.59±3.66 | 57.24±5.97 | 51.36±5.39 | 55.83±4.04 | 56.64±4.36 | 53.75±4.47 | 68.28±6.77 | 81.33±7.72 | 61.31±10.73 |
| | Mean Pooling | 60.77±4.86 | 65.68±6.04 | 60.48±4.87 | 65.96±2.32 | 72.41±3.86 | 64.33±2.27 | 79.62±3.15 | 92.09±1.92 | 76.67±3.49 |
| | ABMIL [27] | 67.31±4.64 | 75.18±5.13 | 66.81±5.22 | 68.96±4.86 | 76.84±4.27 | 68.42±5.45 | 88.89±1.71 | 96.86±0.84 | 87.11±2.44 |
| | DSMIL [33] | 72.76±3.42 | 78.99±3.90 | 72.53±3.41 | 72.29±3.64 | 79.46±2.20 | 72.06±3.54 | 88.89±1.71 | 96.86±0.01 | 87.11±2.44 |
| | CLAM-SB [38] | 72.82±2.68 | 79.47±2.93 | 72.58±2.74 | 71.46±3.82 | 80.09±1.80 | 71.24±4.00 | 88.66±2.17 | 97.58±0.01 | 87.00±2.98 |
| | CLAM-MB [38] | 73.27±3.56 | 80.53±3.76 | 73.25±3.55 | 72.29±2.43 | 78.42±2.75 | 72.24±2.47 | 88.74±1.62 | 97.34±0.01 | 86.83±2.50 |
| | TransMIL [47] | 71.60±4.62 | 78.59±4.86 | 71.21±5.00 | 71.67±3.75 | 78.77±2.92 | 71.56±3.73 | 86.97±1.83 | 96.71±0.01 | 85.01±2.65 |
| | DTFD-MIL [54] | 70.51±5.77 | 77.38±5.26 | 70.33±5.89 | 72.71±2.02 | 79.28±1.81 | 72.66±1.99 | 86.74±1.65 | 96.74±0.71 | 87.06±1.99 |
| | WiKG-MIL [34] | 68.20±3.47 | 75.08±4.66 | 67.98±3.56 | 68.75±3.16 | 75.51±2.16 | 68.59±3.07 | 83.99±1.70 | 95.13±0.70 | 81.54±3.14 |
| | ViLa-MIL [48] | 73.27±5.54 | 80.82±6.41 | 73.24±5.52 | 72.50±3.93 | 77.67±3.12 | 72.35±3.92 | 84.60±1.04 | 95.67±0.70 | 81.42±1.04 |
| | MSCPT [22] | 76.15±3.83 | 84.06±3.02 | 76.13±3.82 | 72.08±6.16 | 78.59±4.21 | 72.06±6.16 | 86.89±0.20 | 96.89±0.70 | 85.33±2.41 |
| | FOCUS [19] | 69.04±3.54 | 74.64±4.29 | 69.00±3.56 | 68.75±4.42 | 75.66±2.86 | 68.47±4.70 | 89.12±1.23 | 97.13±0.46 | 87.43±1.68 |
| | HiVE-MIL | **79.23±2.70** | **87.34±4.08** | **79.09±2.75** | **77.08±3.90** | **84.31±4.22** | **76.80±4.15** | **89.97±0.85** | **98.32±0.45** | **88.18±1.25** |
| | Δ from 2nd-best | (+3.08) | (+3.28) | (+2.96) | (+4.37) | (+4.22) | (+4.14) | (+0.85) | (+0.74) | (+0.75) |
| **CONCH [39]** 1.17M Pathology Image-Text Pairs | Max Pooling | 78.85±1.78 | 87.43±1.69 | 78.82±1.77 | 71.25±2.99 | 78.46±4.53 | 70.91±3.14 | 80.15±4.86 | 91.95±2.76 | 78.11±4.60 |
| | Mean Pooling | 79.55±2.73 | 87.90±2.78 | 79.47±2.74 | 76.67±2.92 | 86.08±4.43 | 76.47±2.81 | 87.74±0.69 | 96.76±0.47 | 86.06±0.46 |
| | ABMIL [27] | 84.30±2.22 | 90.97±0.60 | 84.28±2.21 | 81.04±3.05 | 87.50±5.38 | 80.93±3.04 | 88.43±1.95 | 96.17±0.76 | 86.95±2.33 |
| | DSMIL [33] | 85.83±2.78 | 94.23±1.20 | 85.76±2.84 | 82.08±3.92 | 89.91±5.46 | 81.99±3.89 | 91.95±1.95 | 98.20±0.23 | 90.87±2.00 |
| | CLAM-SB [38] | 85.83±4.25 | 93.19±2.39 | 85.80±4.29 | 82.29±7.42 | 90.70±6.73 | 82.24±7.41 | 92.11±0.52 | 98.17±0.33 | 90.76±0.85 |
| | CLAM-MB [38] | 86.92±3.39 | 94.01±2.16 | 86.91±3.40 | 81.88±4.82 | 90.41±5.14 | 81.84±4.81 | 91.42±1.13 | 98.15±0.22 | 89.96±1.11 |
| | TransMIL [47] | 85.90±3.36 | 93.38±2.11 | 85.88±3.36 | 82.50±5.37 | 89.69±4.54 | 82.38±5.36 | 89.27±2.34 | 97.75±0.69 | 87.66±2.95 |
| | DTFD-MIL [54] | 88.40±3.54 | 95.36±1.52 | 88.37±3.56 | 83.54±3.86 | 91.22±3.39 | 83.48±3.83 | 91.65±1.44 | 97.99±0.09 | 90.38±1.52 |
| | WiKG [34] | 82.24±3.13 | 91.17±1.62 | 82.15±3.21 | 79.58±6.17 | 87.42±6.54 | 79.44±6.39 | 89.73±2.37 | 97.65±0.67 | 87.84±3.12 |
| | ViLa-MIL [48] | 83.08±3.63 | 91.10±2.43 | 83.04±3.64 | 77.08±6.69 | 87.03±8.01 | 76.98±6.73 | 89.27±2.32 | 97.48±0.79 | 87.91±2.88 |
| | MSCPT [22] | 80.06±5.20 | 88.06±6.28 | 79.95±5.24 | 79.79±8.22 | 87.33±6.75 | 79.69±8.21 | 92.03±1.52 | 98.03±0.35 | 90.89±1.94 |
| | FOCUS [22] | 85.32±2.54 | 93.43±1.45 | 85.24±2.60 | 82.50±5.57 | 90.10±4.50 | 82.20±5.77 | 91.57±1.14 | 98.13±0.54 | 90.21±1.37 |
| | HiVE-MIL | **90.39±1.57** | **96.49±0.56** | **90.37±1.58** | **87.29±2.83** | **93.86±0.89** | **87.24±2.85** | **92.34±1.33** | **98.53±0.13** | **91.32±1.68** |
| | Δ from 2nd-best | (+1.99) | (+1.13) | (+2.00) | (+3.75) | (+2.64) | (+3.76) | (+0.23) | (+0.33) | (+0.43) |

baselines, HiVE-MIL achieves significant improvements across all datasets, with gains of up to +4.37% in ACC, +4.22% in AUC, and +4.14% in macro F1 on BRCA, and notable margins on NSCLC (up to +3.27% ACC, +3.28% AUC, +3.26% F1) and RCC (up to +1.07% ACC, +0.85% AUC, +1.64% F1). Even when paired with CONCH, the best pathology VLM to date, HiVE-MIL maintains consistent improvements across all datasets. Traditional MIL methods are based solely on visual features and large WSI labels, making them ineffective in the FSWC setting. Existing VLM-MIL methods, including multi-scale approaches, fail to capture hierarchical interactions across scales and do not effectively align modalities within the same scale. In contrast, HiVE-MIL explicitly models cross-scale hierarchies and intra-scale modality alignments, leading to superior performance.

## 4.3 Robustness in Few-Shot Scenarios

Figure 3 shows the performance of HiVE-MIL in 4-, 8-, and 16-shot settings across datasets and pathology-specific VLMs. Even with extremely limited supervision, HiVE-MIL consistently outperforms existing baselines. For example, the highest observed performance gains are +6.81% and +8.57% on NSCLC at 4- and 8-shot settings with Quilt-Net, and +3.48% and +3.83% on BRCA with QuiltNet. The consistent improvements across diverse scenarios reflect the method's effectiveness in FSWC settings.

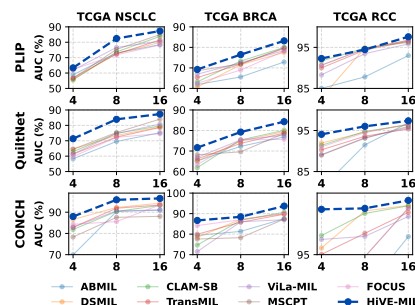

Figure 3: Few-shot robustness.

## 4.4 Hierarchical Text Semantic Alignment

Figure 4 evaluates the relationship between hierarchical textual semantic consistency and classification performance, analyzing how effectively HiVE-MIL aligns parent–child structures across low- and high-scale textual descriptions. For each low-scale patch, we retrieve its top-$K$ ($K$=2) most similar low-scale texts (parents). We then identify the high-scale patches linked to the same low-scale patch and check whether their most similar high-scale text (child) corresponds to any child text associated with the retrieved parent texts. A *hit* is recorded if such a match is found (see the Appendix F for more details). This process is repeated across all low-scale patches in the test WSIs and the overall score is reported as Hit Ratio@2 (x-axis). Among the evaluated variants (PLIP and QuiltNet, 16-shot), the bidirectional (*Bi.*) HiVE-MIL consistently achieves the highest Hit Ratio, outperforming both unidirectional (*Uni.*) and no-interaction (*No.*) variants. The strong correlation between Hit Ratio and Macro F1 underscores the effectiveness of bidirectional message passing in preserving hierarchical semantics across scales.

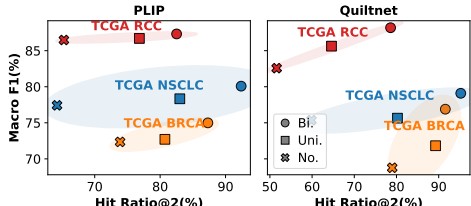

Figure 4: Performance of Hit Ratio@2 and Macro F1 (16-shot).

## 4.5 Interpretability Analysis

To assess interpretability, we visualize how HiVE-MIL aligns visual patches with class-level semantics at each scale using visual textual similarity scores. We sample a WSI from the IDC class in the BRCA dataset and identify, at each scale, the patch with the highest similarity to the class text, referred to as the *Anchor* (Figure 5). We then select *Positive* patches with text distributions most similar to the Anchor, and *Negative* patches with the most dissimilar distributions. The anchor and positive patches show similar morphological patterns, whereas the negative patches differ clearly in structure. We also show

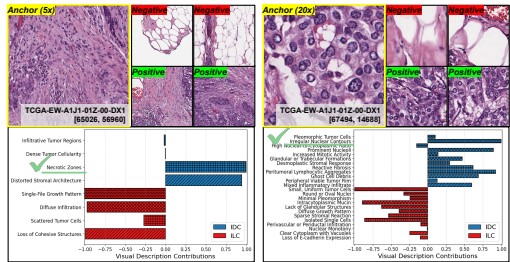

Figure 5: Low- and high-scale patches with highest (*Anchor*, *Positive*) and lowest (*Negative*) similarity to the WSI label (BRCA, CONCH, 16-shot).

that IDC-related text classes have higher probabilities than ILC-related ones for each Anchor. This supports HiVE-MIL's IDC prediction for WSI and provides interpretable evidence based on the description of the contributing text.

## 4.6 Ablation Studies

**Variants of HTCL.** We evaluate three variants of the hierarchical text contrastive loss (HTCL), each defined by a distinct strategy for selecting anchors, positives, and negatives. In the *Share-Parent* variant, the anchor is the mean embedding of high-scale texts that share the same low-scale parent ($\bar{\mathbf{T}}_i^{(h)}$), and the query is a low-scale embedding ($\mathbf{T}_j^{(l)}$). Positives and negatives are defined as $\mathcal{P}_i = \{j \mid \text{Parent}(i) = j\}$ and $\mathcal{N}_i = \{j \mid \text{Parent}(i) \neq j\}$. The *Instance-Wise* variant uses each high-scale embedding $\mathbf{T}_i^{(h)}$ as the anchor, with positive and negative sets identical to those in *Share-Parent*. Our proposed *Class-Wise* variant (Section 3.4.2) constructs positive and negative sets based on shared class labels on all scales, allowing supervision at the class level. As shown in Table 2, *Class-Wise* consistently achieves the best performance across all settings. Moreover, all HTCL variants outperform the *No-Contrastive* baseline that uses only cross-entropy loss, highlighting the benefit of contrastive supervision for hierarchical text semantic alignment. Please refer to the Appendix J.1 for details on the HTCL variants.

**Effects of Module Components.** Table 3 presents an ablation study evaluating the contributions of TGDF, HHG, and HTCL. The full model (d), which integrates all three components, achieves the highest accuracy and Macro F1 across all datasets, highlighting their complementary effects. Removing HTCL (c) leads to a performance drop on BRCA, with a 2.04% decrease in Macro F1, emphasizing the importance of enforcing textual semantic consistency across scales. Further

Table 2: HTCL Variants (PLIP, 16-shot).

|  | TCGA NSCLC | | TCGA BRCA | |
|---|---|---|---|---|
|  | ACC | Macro F1 | ACC | Macro F1 |
| *No-Contrastive* | 78.14 ±3.55 | 78.11 ±3.54 | 73.96 ±4.42 | 73.81 ±4.46 |
| *Share-Parent* | 78.46 ±3.71 | 78.37 ±3.71 | 74.17 ±3.45 | 73.83 ±3.41 |
| *Instance-Wise* | 78.59 ±3.99 | 78.55 ±4.01 | 75.00 ±2.38 | 74.78 ±2.33 |
| **Class-Wise (Ours)** | **80.13** ±4.73 | **80.08** ±4.73 | **75.21** ±3.51 | **74.99** ±3.67 |

Table 3: Module ablation (QuiltNet, 16-shot).

| Row | TGDF | HHG | HTCL | TCGA NSCLC | | TCGA BRCA | | TCGA RCC | |
|---|---|---|---|---|---|---|---|---|---|
|  |  |  |  | ACC | Macro F1 | ACC | Macro F1 | ACC | Macro F1 |
| (a) | ✗ | ✗ | ✗ | 74.73 ±4.23 | 73.65 ±3.04 | 69.38 ±5.81 | 69.24 ±5.74 | 86.23 ±0.42 | 84.98 ±1.12 |
| (b) | ✓ | ✗ | ✗ | 77.01 ±2.71 | 76.80 ±2.98 | 73.13 ±3.39 | 72.75 ±3.56 | 87.36 ±1.80 | 85.21 ±1.92 |
| (c) | ✓ | ✓ | ✗ | 78.33 ±3.82 | 78.27 ±3.78 | 75.17 ±4.92 | 74.76 ±4.23 | 88.82 ±0.66 | 86.85 ±0.78 |
| (d) | ✓ | ✓ | ✓ | **79.23** ±2.70 | **79.09** ±2.75 | **77.08** ±3.90 | **76.80** ±4.15 | **89.97** ±0.85 | **88.18** ±1.25 |

removing HHG (b) results in an additional 1.64% drop in Macro F1 on RCC compared to (c), indicating the critical role of hierarchical message passing in capturing multi-scale relationships. The configuration without any of the modules (a) consistently yields the lowest performance, confirming the necessity of each component. Overall, these results demonstrate that TGDF, HHG, and HTCL are jointly essential for effective representation learning in few-shot WSI classification. Please refer to the Appendix J.2 for descriptions of the module component ablations.

**Effects of Hierarchical Aggregator.** As shown in Figure 6, the Modality-Scale Attention (*MSA*) module consistently outperforms all baselines across the evaluation metrics, including Modality-Aware Attention (*MAA*), Scale-Aware Attention (*SAA*), generic Attention (*Attn.*), and GraphSAGE (*SAGE*). *MSA* explicitly models both the

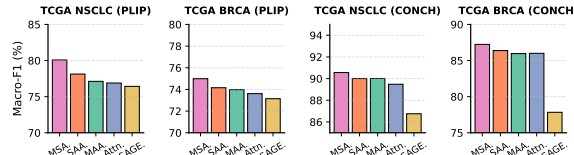

Figure 6: Comparison of hierarchical aggregator methods (16-shot).

modality and scale at the edge level, enabling more effective interaction throughout the hierarchy. In contrast, the other variants omit modality, scale, or both, resulting in a performance drop of 1–3%. This highlights the importance of jointly modeling modality-specific and scale-aware information. *SAGE* performs the worst, as it applies uniform intra-scale message passing without scale adaptation, unlike other variants that incorporate scale-specific design. Details on hierarchical aggregator methods are provided in the Appendix J.3.

**Further Ablations.** We provide additional ablation studies in Appendix M and hyperparameter sensitivity analyses in Appendix N. The findings indicate that HiVE-MIL is robust to a range of hyperparameter settings. Additionally, we report FLOPs, inference time, and maximum GPU memory usage in Appendix O, demonstrating that although our method incurs moderate computational overhead compared to the baselines, it remains efficient and delivers competitive performance.

# 5 Discussion

**Conclusion.** We propose HiVE-MIL, a hierarchical vision-language MIL framework that models hierarchical dependencies and intra-scale multimodal alignments through a unified hierarchical heterogeneous graph. Hierarchical edges enhance contextual understanding by message passing across scales, while intra-scale links ensure semantic consistency across modalities. These designs, combined with text-guided filtering and hierarchical contrastive loss, enable robust learning under few-shot supervision. HiVE-MIL consistently outperforms MIL and VLM-MIL baselines across three TCGA datasets, offering an effective solution to scale-aware, multimodal WSI classification.

**Limitations and Future Work.** TGDF currently relies on non-learnable, similarity-based thresholds defined per WSI, which may limit generalization across datasets or backbones. As future work, we plan to explore adaptive, learnable filtering mechanisms to enhance robustness and transferability. Additionally, the Hit Ratio metric assumes the correctness of LLM-generated parent–child text structures. We plan to incorporate expert human evaluation to validate these hierarchies and to assess the alignment between the highest text description contribution with the corresponding patch.

**Acknowledgements.** This work was supported by the National Research Foundation of Korea (NRF) grant funded by the Korean government (MSIT) under grant numbers RS-2022-NR068758 and NRF-2025-25432820. We are also deeply grateful for the generous support provided by the Seegene Medical Foundation in South Korea.

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

# Appendix for
# Few-Shot Learning from Gigapixel Images via Hierarchical Vision-Language Alignment and Modeling

## A  Dataset and Input Construction

### A.1  Dataset Descriptions

We use three public WSI datasets from The Cancer Genome Atlas (TCGA): NSCLC, BRCA, and RCC.[4] Each slide undergoes Otsu's thresholding to remove background regions and is normalized for stain variation. Patches are extracted at two scales: $256 \times 256$ pixels at $5\times$ (low scale) and $20\times$ (high scale). Table 4 summarizes the number of slides and patches extracted for each subtype.

Table 4: Summary of TCGA datasets used in this study.

| Dataset | Subtype (Abbr.) | # Slides | # Patches ($5\times$) | # Patches ($20\times$) |
|---|---|---|---|---|
| TCGA NSCLC | Lung Adenocarcinoma (LUAD) | 531 | 452,664 | 6,917,186 |
| | Lung Squamous Cell Carcinoma (LUSC) | 511 | 424,008 | 6,451,978 |
| TCGA BRCA | Invasive Ductal Carcinoma (IDC) | 844 | 621,958 | 9,276,899 |
| | Invasive Lobular Carcinoma (ILC) | 211 | 140,974 | 2,092,908 |
| TCGA RCC | Clear Cell Renal Cell Carcinoma (CCRCC) | 455 | 424,338 | 6,487,063 |
| | Papillary Renal Cell Carcinoma (PRCC) | 296 | 258,288 | 3,909,258 |
| | Chromophobe Renal Cell Carcinoma (CHRCC) | 121 | 113,148 | 1,725,880 |

Following [48], we split each dataset into training, validation, and test sets using a 4:3:3 ratio. For a few-shot evaluation, we repeat the random-splitting process five times and report averaged results. Due to the class imbalance in the TCGA BRCA (approximately 20:80), we balance the test set during sampling to avoid misleading final metric results. All details regarding the data splits used are provided in the *splits* directory of our GitHub repository (linked in the Abstract).

### A.2  Hierarchical Patch Extraction

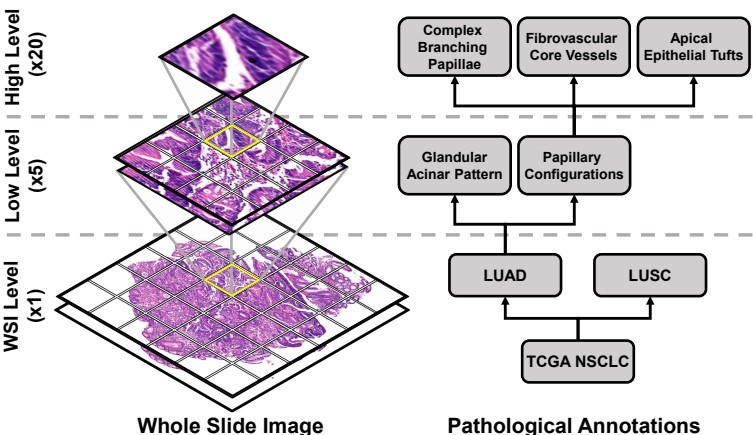

Figure 7: Illustrative example of hierarchical visual patches and hierarchical texts within a WSI.

Existing MIL methods [33, 48, 22, 41] typically process patch features at different scales independently, without preserving any hierarchical structure between them. This design hinders the ability

---

[4] https://portal.gdc.cancer.gov/

to model structured cross-scale interactions, which are essential in tasks such as WSI classification, where coarse tissue-level patterns and fine-grained cellular features are often interdependent. To address this, we propose a hierarchical extraction pipeline with an explicit 1-to-16 mapping across scales: each low-scale ($5\times$) patch corresponds to a spatial region at the high scale ($20\times$), which is uniformly subdivided into a fixed $4 \times 4$ grid, yielding 16 high-scale patches. This structured mapping ensures that each high-scale patch can be directly and uniquely *traced back* to its low-scale parent, enabling consistent alignment and hierarchical reasoning across scales.

To facilitate downstream processing, we flatten the two-level indices $(n, m)$, where $n$ denotes the low-scale patch index and $m \in \{0, \ldots, 15\}$ denotes the position of the grid within its corresponding high-scale region, into a single index $r = 16n + m$. This indexing convention enables the effective retrieval of *parent-child* relationships and is used throughout all scale-sensitive components. It forms the basis of our multi-scale vision-language alignment framework, supporting semantic propagation across scales while preserving spatial coherence within the WSI.

### A.3 Hierarchical Textual Prompt Construction

We present the hierarchical textual descriptions used in our study to guide visual-language alignment in HiVE-MIL. The final version of the text is generated using **GPT-4o** [1], although we also explore variants generated by other LLMs and demonstrate that HiVE-MIL's performance is robust to the choice of LLM (see Appendix M.2). These descriptions capture representative morphological features at two levels: coarse tissue-level patterns at the $5\times$ scale and fine-grained cellular structures at the $20\times$ scale. Each coarse-level feature (four in total) serves as a *parent*, linked to three fine-level *child* features, forming a structured hierarchy. These hierarchical descriptions are used as textual input in our HiVE-MIL framework.

## TCGA NSCLC (LUAD) (5× → 20×)

**Glandular Acinar Patterns**: Well-circumscribed acinar structures composed of atypical cuboidal to columnar epithelial cells, often containing intraluminal mucin and exhibiting mild nuclear pleomorphism.

- *Irregular Gland Outlines*: Angulated, non-uniform glandular profiles with irregular lumina, nuclear crowding, and focal mucin, indicating infiltrative glandular transformation.
- *Nuclear Hyperchromasia*: Darkly stained, enlarged nuclei with coarse chromatin and prominent nucleoli, often accompanied by atypical mitotic figures.
- *Fibrotic Stromal Response*: Dense fibrous stroma with activated fibroblasts and peritumoral lymphocytic infiltration, reflecting the desmoplastic reaction around the malignant glands.

**Papillary Configurations**: Branching papillary structures supported by delicate fibrovascular cores, lined by stratified columnar tumor cells with nuclear pseudo-stratification and occasional apical tufting.

- *Complex Branching Papillae*: Elaborately branched papillary fronds lined by multilayered atypical cells, often with central necrosis and early signs of stromal infiltration.
- *Fibrovascular Core Vessels*: Central capillaries within papillary stalks, surrounded by scant inflammatory cells and sometimes showing endothelial proliferation or changes in the basement membrane.
- *Apical Epithelial Tufts*: Protruding apical structures formed by columnar tumor cells with eosinophilic cytoplasm and low-grade nuclear stratification, occasionally secreting mucin.

**Lepidic Growth**: Tumor cells spreading along intact alveolar septa without architectural distortion, preserving pulmonary parenchyma and lacking stromal invasion.

- *Flat Monolayer Proliferation*: A single layer of atypical cells that line the alveolar walls, maintaining the underlying architecture with subtle nuclear atypia and minimal stromal response.
- *Preserved Alveolar Architecture*: Retention of the alveolar septa and capillary framework despite tumor cell proliferation, often accompanied by mild inflammation.
- *Mucinous Secretions*: Extracellular mucin accumulation within alveolar spaces, bordered by flattened tumor cells, suggesting early mucin-producing differentiation.

**Solid Sheets with Mucin Production**: Compact tumor cell sheets without glandular formation, exhibiting cytoplasmic mucin vacuoles, nuclear pleomorphism, and high nuclear-to-cytoplasmic ratios.

- *Coalescent Cell Clusters*: Irregularly bordered nests or aggregates of poorly differentiated tumor cells with hyperchromatic nuclei and scant glandular features.
- *Intracellular Mucin Pools*: Cytoplasmic mucin displacing the nucleus to the periphery, creating signet-ring-like cells typical of mucinous adenocarcinoma variants.
- *Central Necrotic Foci*: Necrotic zones within solid tumor areas, surrounded by viable malignant cells, showing karyorrhexis, inflammation, and early cavitation.

## TCGA NSCLC (LUSC) (5× → 20×)

**Keratin Pearls**: Concentric layers of eosinophilic keratinized material encircled by malignant squamous cells, hallmark of well-differentiated squamous carcinoma.

- *Concentric Lamellae*: Onion-skin–like layers of compact keratin, indicating advanced squamous maturation in well-differentiated tumors.
- *Central Keratin Accumulation*: Dense eosinophilic keratin material in the center of tumor nests, displacing surrounding malignant cells outward.
- *Peripheral Reactive Stroma*: Fibroblastic and inflammatory stromal response surrounding keratinizing nests, often with multinucleated giant cells.

**Intercellular Bridges**: Cytoplasmic connections between adjacent tumor cells, representing retained desmosomal junctions typical of squamous differentiation.

- *Desmosomal Thickenings*: Pronounced desmosomal plaques visible at tumor cell junctions, reinforcing epithelial cohesion in squamous cells.
- *Bridging Spines*: Elongated cytoplasmic projections maintaining intercellular contacts, characteristic of squamous cell architecture.
- *Intercellular Gaps*: Narrow spaces between adjacent tumor cells with intact desmosomal attachments, allowing minimal extracellular fluid passage.

**Dyskeratotic Cells**: Isolated eosinophilic tumor cells undergoing premature keratinization, appearing as dense, glassy bodies within cell clusters.

- *Premature Keratin Accumulation*: Early keratin buildup within individual cells, causing nuclear condensation and cytoplasmic eosinophilia in dyskeratotic zones.
- *Eccentric Hyperchromatic Nuclei*: Peripheral, dark-staining nuclei compressed by cytoplasmic keratin, showing irregular contours and coarse chromatin.
- *Focal Cell Lysis*: Localized necrosis with keratin extrusion and surrounding inflammatory infiltrates, occurring within tumor nests.

**Squamous Eddies**: Swirling arrangements of keratinizing squamous cells, forming eddy-like patterns often seen in keratinizing regions.

- *Spiralized Cellular Streams*: Corkscrew-like arrangements of tumor cells showing organized squamous maturation with partial keratinization.
- *Vortex-Like Whorls*: Dense, spiral patterns of malignant squamous cells with central keratinization and peripheral nuclear reorientation.
- *Peripheral Flattening*: Peripheral squamous cells becoming flattened along the edges of nests, demarcating mature keratinizing foci.

## TCGA BRCA (IDC) (5× → 20×)

**Infiltrative Tumor Regions**: At 5×, irregularly shaped, ill-defined tumor regions invade surrounding tissue, displacing or distorting adjacent structures.

- *Pleomorphic Tumor Cells*: At 20×, tumor cells display marked variation in size and shape, with irregular contours and disorganized architecture.
- *Irregular Nuclear Contours*: Nuclei show jagged, wrinkled borders, with coarse chromatin and frequent mitotic figures.
- *High Nuclear-to-Cytoplasmic Ratio*: Tumor cells have large hyperchromatic nuclei and scant cytoplasm, reflecting aggressive cellular proliferation.

**Dense Tumor Cellularity**: Low-power view reveals solid nests or trabecular cords of tumor cells occupying large portions of the parenchyma.

- *Prominent Nucleoli*: Nuclei often feature conspicuous eosinophilic nucleoli, characteristic of high-grade IDC.
- *Increased Mitotic Activity*: Numerous mitoses, including abnormal forms, are visible, particularly at invasive fronts.
- *Glandular or Trabecular Formations*: Tumor architecture includes malformed gland-like ducts or linear trabeculae amidst fibrotic stroma.

**Necrotic Zones**: Focal areas of necrosis, sometimes centrally located within tumor nests, give rise to ghost cell zones and debris.

- *Ghost Cell Debris*: Necrotic areas contain pyknotic nuclei, faded cytoplasm, and fragmented cell remnants.
- *Peripheral Viable Tumor Rim*: Viable tumor cells form a rim around necrosis, often with hyperchromatic nuclei and mitotic activity.
- *Mixed Inflammatory Infiltrate*: Neutrophils and macrophages infiltrate around necrotic zones, indicating the tumor-host interaction.

**Distorted Stromal Architecture**: Stroma shows reactive changes and desmoplasia due to infiltrative and space-occupying tumor growth.

- *Desmoplastic Stromal Response*: Fibroblast-rich, fibrotic stroma surrounds tumor nests and glandular elements.
- *Reactive Fibrosis*: Dense collagen bundles with stromal retraction and scattered lymphocytes surround invasive regions.
- *Peritumoral Lymphocytic Aggregates*: Clusters of immune cells at the tumor-stroma borders suggest the host response to invasion.

## TCGA BRCA (ILC) (5× → 20×)

**Single-File Growth Pattern**: At 5×, tumor cells infiltrate in linear rows between stromal fibers, forming the classic Indian file architecture.

- *Small, Uniform Tumor Cells*: At 20×, tumor cells are monomorphic, with small round-to-oval nuclei and inconspicuous nucleoli.
- *Round or Oval Nuclei*: Nuclear morphology is bland, with smooth contours and fine chromatin.
- *Minimal Pleomorphism*: Cellular features are uniform, with very limited variability in size, shape, or staining.

**Diffuse Infiltration**: Tumor cells are widely dispersed across the stroma, lacking a mass-forming architecture.

- *Intracytoplasmic Mucin*: Clear cytoplasmic vacuoles displace nuclei, creating a targetoid or signet-ring–like appearance.
- *Lack of Glandular Structures*: No lumen formation or epithelial polarity is observed, which differentiates ILC from IDC.
- *Diffuse Growth Pattern*: Tumor spreads diffusely through the stroma, often preserving native tissue landmarks.

**Scattered Tumor Cells**: Neoplastic cells appear loosely distributed, often without clustering or clear nest formation.

- *Sparse Stromal Reaction*: Fibrous stroma is loose and minimally reactive, in contrast to the desmoplasia in IDC.
- *Isolated Single Cells*: Tumor cells often appear as single entities, lacking intercellular adhesion or junctions.
- *Perivascular or Periductal Infiltration*: Tumor cells wrap around existing ducts and vessels without destruction, maintaining normal tissue outlines.

**Loss of Cohesive Structures**: Absence of glandular or trabecular patterns, with minimal distortion of pre-existing tissue architecture.

- *Nuclear Monotony*: Cells exhibit a uniform nuclear size, shape, and chromatin, reflecting low-grade morphology.
- *Clear Cytoplasm with Vacuoles*: Cytoplasm appears pale with discrete vacuoles, often mistaken for benign tissue.
- *Loss of E-cadherin Expression*: Absence of cell adhesion molecules explains discohesive behavior, visible through architectural disarray.

## TCGA RCC (CCRCC) (5× → 20×)

**Clear Cytoplasm Dominance**: At 5×, tumor cells appear in sheets or nests with optically clear cytoplasm due to lipid/glycogen, bounded by crisp membranes.

- *Intracytoplasmic Glycogen/Lipid Accumulation*: At 20×, cytoplasm is vacuolated and optically clear due to the lipid or glycogen content.
- *Peripheral Nuclear Displacement*: The nuclei are eccentrically located and pushed to the periphery of the cell by abundant cytoplasm.
- *Fine Cell Membrane Borders*: Well-demarcated cell borders are easily identifiable at high magnification.

**Alveolar-Nested Architecture**: Low power reveals pseudo-alveolar structures formed by small tumor nests and intervening thin fibrovascular septa.

- *Small Tumor Nests*: Compact, round nests of tumor cells mimic alveolar units, often surrounded by fine capillaries.
- *Intervening Thin Fibrovascular Septa*: Nests are separated by delicate fibrovascular septa lined by flat endothelial cells.
- *Lack of Papillary Structures*: No fibrovascular cores or true papillae are evident, which helps to distinguish the subtype.

**Prominent Sinusoidal Vasculature**: Numerous sinusoidal capillaries form arborizing patterns that wrap around tumor clusters.

- *Rich Capillary Networks*: 20× shows dense, branching vascular beds extending between and around tumor clusters.
- *Endothelial Wrapping*: Capillaries encase the nests with endothelial cells forming tight boundaries.
- *Erythrocyte-Filled Lumina*: Sinusoidal spaces frequently appear engorged with red blood cells.

**Minimal Nuclear Atypia**: Tumor nuclei show minimal pleomorphism, with low-grade features distributed evenly across nests.

- *Low Nuclear Grade*: Nuclei are round with uniform chromatin, corresponding to Fuhrman grade I–II.
- *Inconspicuous Nucleoli*: Nucleoli are small or absent under 20×, suggesting low proliferative activity.
- *Uniform Chromatin*: Evenly distributed chromatin supports low-grade tumor morphology.

## TCGA RCC (CHRCC) (5× → 20×)

**Pale Eosinophilic Cytoplasm**: At 5×, tumor cells exhibit pale pink cytoplasm with central nuclei and minimal architectural variation.

- *Prominent Perinuclear Halo*: Clear perinuclear zones caused by microvesicular cytoplasm dominate the cell morphology.
- *Reticulated Cytoplasm*: Fine vesicle-like reticulations are visible in cytoplasm under higher magnification.
- *Dense Cell Borders*: Polygonal tumor cells have well-defined borders and cytoplasmic outlines.

**Solid Sheet Growth Pattern**: Tumor grows in broad, cohesive sheets with minimal stromal interruption or patterning.

- *Broad Cell Plates*: Cells form expansive and cohesive units with minimal architectural disruption.
- *Lack of Fibrovascular Core*: The growth pattern lacks central fibrovascular structures, reinforcing a solid architecture.
- *Sparse Mitoses*: Few mitotic figures are observed, suggesting low-grade proliferative activity.

**Perinuclear Clearing**: Halo-like clearing around nuclei imparts a distinct plant-cell morphology.

- *Clear Perinuclear Cytoplasmic Zones*: Large halos or perinuclear clearing disrupts cytoplasmic uniformity.
- *Irregular Nuclei*: Nuclear contours are wrinkled or raisinoid in appearance.
- *Binucleation*: Multiple nuclei within a single cell are commonly observed.

**Plant-like or Mosaic Growth**: Geographic arrangements of cells create a tiled or mosaic-like appearance under low magnification.

- *Geographic Cell Grouping*: Tumor cells form clustered patches resembling tiles or islands.
- *Peripheral Cytoplasmic Accentuations*: Borders are thickened or accentuated at the cell periphery.
- *Eosinophilic Granularity*: Cytoplasm contains fine, pink-staining granules.

## TCGA RCC (PRCC) (5× → 20×)

**Papillary/Trabecular Architecture**: At 5×, tumor growth includes true papillae and trabecular structures with alternating cords and sheets.

- *True Papillae*: Well-formed fibrovascular cores lined by a single or pseudo-stratified tumor epithelium are present.
- *Pseudopapillary Areas*: Incomplete or collapsed papillary structures lacking central cores are seen.
- *Trabecular Slits*: Linear cords of tumor cells form slit-like spaces within loose fibrotic stroma.

**Foamy Macrophage Aggregates**: Pale yellow zones containing lipid-laden macrophages are seen within tumor stroma and papillae.

- *Intraluminal Clusters*: Macrophages aggregate in luminal spaces or within papillary cores.
- *Vacuolated Cytoplasm*: Lipid content gives macrophages a foamy, vacuolated appearance.
- *Hemosiderin Pigmentation*: Golden-brown pigment granules are deposited within macrophage cytoplasm.

**Pseudostratified Tumor Epithelium**: Papillae are lined by tumor cells with crowded, elongated nuclei mimicking stratification.

- *High Nuclear Crowding*: Elongated nuclei densely crowd near the apical surface of epithelial layers.
- *Hyperchromatic Nuclei*: Nuclei are darkly stained, often irregular in shape.
- *Mitotic Figures*: Frequent mitotic activity is visible, especially in higher-grade cases.

**Psammoma Body Formation**: Calcified concentric structures are visible in the cores or adjacent stroma under low magnification.

- *Concentric Calcification*: The bodies of psammoma appear as round layered calcifications in the stroma.
- *Stromal Mineralization*: Stromal areas show scattered calcific debris.
- *Associated Necrosis*: Focal necrotic zones are seen near the papillae or within the stroma.

# B  Validation of LLM-Generated Descriptions

While the descriptions generated by the LLM have not yet been verified by pathologists, we address this limitation through a series of alternative validation strategies designed to assess their class-discriminative relevance and semantic reliability in capturing morphology- and concept-level information.

## B.1  Class-Discriminative Relevance

We evaluate the *faithfulness* of the generated descriptions using the *LLM-as-a-Judge* [16], where LLMs are prompted to infer class labels based solely on the generated descriptions. We employ two evaluators: GPT-4o and GPT o3, to assess descriptions generated by several LLMs: Deep Seek R1 [18], Grok 3 [53], Gemini 2.5 Pro [3], and GPT-4o [1]. Although not perfect, the number of correct predictions is still consistently high, suggesting that the generated text is generally reliable while capturing class-discriminative signals.

Table 5: Class-discriminative relevance (LLM-as-a-Judge).

| Generator / Evaluator | Total Descriptions (#5× / #20×) | GPT-4o (TCGA NSCLC) | GPT-4o (TCGA BRCA) | GPT o3 (TCGA NSCLC) | GPT o3 (TCGA BRCA) |
|---|---|---|---|---|---|
| DeepSeek R1 [18] | 32 (8/24) | 28 (8/20) | 28 (8/20) | 31 (8/23) | 28 (8/20) |
| Grok 3 [53] | 32 (8/24) | 28 (7/21) | 28 (8/20) | 32 (8/24) | 21 (8/23) |
| Gemini 2.5 Pro [53] | 32 (8/24) | 30 (8/22) | 29 (7/22) | 30 (7/23) | 32 (8/24) |
| GPT-4o [1] | 32 (8/24) | 30 (8/22) | 30 (7/23) | 31 (8/23) | 32 (8/24) |
| **Total** | 128 (32/96) | 116 (31/85) | 114 (29/85) | 124 (31/93) | 113 (32/89) |

## B.2  Morphology- and Concept-Level Semantic Validation

Table 6 reports intra- and inter-category text similarity scores, where *intra* denotes similarities among descriptions of the same morphology or concept, and *inter* denotes similarities across different ones. Similarity is computed separately at each scale and jointly across both scales using the CONCH text encoder [39] and cosine similarity. These results demonstrate that the descriptions encode highly discriminative and consistent semantics across both morphological groups and scales, and across different LLMs.

Table 6: Intra- and inter-category text similarity scores. *Intra* denotes similarities within the same morphology or concept across scales, and *inter* denotes similarities across different ones.

| LLM | TCGA NSCLC | | | TCGA BRCA | | |
|---|---|---|---|---|---|---|
| | 5× (intra/inter) | 20× (intra/inter) | 5×,20× (intra/inter) | 5× (intra/inter) | 20× (intra/inter) | 5×,20× (intra/inter) |
| DeepSeek R1 [18] | 1.000 (0.157) | 0.515 (0.164) | 0.456 (0.158) | 1.000 (0.165) | 0.438 (0.141) | 0.412 (0.138) |
| Grok 3 [53] | 1.000 (0.123) | 0.511 (0.170) | 0.434 (0.160) | 1.000 (0.186) | 0.490 (0.214) | 0.415 (0.212) |
| Gemini 2.5 Pro [3] | 1.000 (0.111) | 0.558 (0.165) | 0.478 (0.150) | 1.000 (0.206) | 0.491 (0.127) | 0.426 (0.142) |
| GPT-4o [1] | 1.000 (0.066) | 0.516 (0.114) | 0.454 (0.109) | 1.000 (0.163) | 0.477 (0.108) | 0.413 (0.125) |

We further validate the performance of HiVE-MIL using descriptions generated by different LLMs, which are provided in Appendix M.2.

# C  Notations

Table 7: Summary of the notations. For simplicity, we describe the notation based on a head-branch.

| Notation | Definition |
|---|---|
| **General Notations** | |
| $N$ | Number of low-scale ($5\times$) patches per WSI |
| $M$ | Number of high-scale ($20\times$) patches per low-scale patch (default: 16) |
| $R$ | Total number of high-scale patches: $R = N \cdot M$ |
| $O$ | Number of low-scale text prompts per class |
| $K$ | Number of high-scale text prompts per low-scale prompt |
| $S$ | Total number of high-scale text prompts: $S = O \cdot K$ |
| $D$ | Dimensionality of visual/textual embeddings |
| $\gamma$ | Logit scaling factor from the pretrained VLM |
| **Visual and Textual Features** | |
| $z_n^{(l)}$ | Visual feature of low-scale patch $n$, $z_n^{(l)} \in \mathbb{R}^D$ |
| $z_{n,m}^{(h)}$ | Visual feature of high-scale patch $m$ in low-scale patch $n$ |
| $z_r^{(h)}$ | Flattened high-scale visual feature where $r = (n, m)$ |
| $Z^{(l)}$ | Low-scale patch features, $Z^{(l)} \in \mathbb{R}^{N \times D}$ |
| $Z^{(h)}$ | High-scale patch features, $Z^{(h)} \in \mathbb{R}^{R \times D}$ |
| $t_o^{(l)}$ | Prompt embedding for low-scale text $o$ |
| $t_{o,k}^{(h)}$ | Prompt embedding for high-scale text $k$ under low-scale text $o$ |
| $e_o^{(l)}$ | Encoded feature of $t_o^{(l)}$, $e_o^{(l)} \in \mathbb{R}^D$ |
| $e_{o,k}^{(h)}$ | Encoded feature of $t_{o,k}^{(h)}$, $e_{o,k}^{(h)} \in \mathbb{R}^D$ |
| $e_s^{(h)}$ | Flattened high-scale text feature where $s = (o, k)$ |
| $E^{(l)}$ | Low-scale text features, $E^{(l)} \in \mathbb{R}^{O \times D}$ |
| $E^{(h)}$ | High-scale text features, $E^{(h)} \in \mathbb{R}^{S \times D}$ |
| $\mathbf{X}^{(s)}$ | Patch feature matrix at scale $s \in \{l, h\}$ |
| $\mathbf{T}^{(s)}$ | Text feature matrix at scale $s \in \{l, h\}$ |
| **Similarity Matrices and Filtering** | |
| $S^{(l)}$ | Low-scale similarity matrix, $S^{(l)} \in \mathbb{R}^{N \times O}$ |
| $S^{(h)}$ | High-scale similarity matrix, $S^{(h)} \in \mathbb{R}^{R \times S}$ |
| $S_{\text{filtered}}^{(l)}$ | Binary mask for low-scale similarity: above $\mu_o + \alpha \cdot \sigma_o$ |
| $S_{\text{masked}}^{(h)}$ | High-scale similarity masked by low-scale filtering: $S^{(h)} \cdot S_{\text{filtered}}^{(l)}$ |
| $S_{\text{filtered}}^{(h)}$ | Final filtered high-scale similarity: above $\mu_s + \alpha \cdot \sigma_s$ |
| $\text{logits}^{(s)}$ | Logits computed by patch-text alignment at scale $s$ |
| **Graph Definitions** | |
| $\mathcal{G}_{\text{HHG}}$ | Hierarchical Heterogeneous Graph |
| $\mathcal{V}$ | Node set of $\mathcal{G}_{\text{HHG}}$ |
| $\mathcal{E}$ | Edge set of $\mathcal{G}_{\text{HHG}}$, $\mathcal{E} = \mathcal{E}^{\text{intra}} \cup \mathcal{E}^{\text{hier}}$ |
| $\mathcal{T}$ | Node types: $\{\text{img}^{(l)}, \text{img}^{(h)}, \text{text}^{(l)}, \text{text}^{(h)}\}$ |
| $\mathcal{R}$ | Edge relation types: $\{\text{intra}^{(l)}, \text{intra}^{(h)}, \text{hier}^{(\text{img})}, \text{hier}^{(\text{text})}\}$ |
| **Aggregation and Prediction** | |
| $h_v^{\text{intra}}$ | Intra-scale aggregated node feature for node $v$ |
| $h_v^{\text{hier}}$ | Hierarchical aggregated node feature via MSA |
| $h_v$ | Final node feature: $h_v = h_v^{\text{intra}} + h_v^{\text{hier}}$ |
| **Loss Terms** | |
| $\mathcal{L}_{\text{CE}}$ | Cross-entropy loss |
| $\mathcal{L}_{\text{HTCL}}$ | Hierarchical Text Contrastive Loss |
| **Hyperparameters** | |
| $\alpha$ | Threshold sensitivity for TGDF filtering |
| $\lambda$ | Weighting factor for $\mathcal{L}_{\text{HTCL}}$ |
| $\text{Top-}k^{(s)}(\cdot)$ | Top-$k$ similarity scores at scale $s$ |

# D Text-Guided Dynamic Filtering Pseudocode

---

**Algorithm 1** Text-Guided Dynamic Filtering (TGDF)

---

**Require:** Low-scale patch features $Z^{(l)} \in \mathbb{R}^{N \times D}$, low-scale text features $E^{(l)} \in \mathbb{R}^{O \times D}$, high-scale patch features $Z^{(h)} \in \mathbb{R}^{R \times D}$, high-scale text features $E^{(h)} \in \mathbb{R}^{S \times D}$, sensitivity parameter $\alpha$

**Ensure:** Filtered similarity matrices $S_{\text{filtered}}^{(l)} \in \mathbb{R}^{N \times O}$, $S_{\text{filtered}}^{(h)} \in \mathbb{R}^{R \times S}$

1: **Stage 1: Low-scale Filtering**
2: $S^{(l)} \leftarrow \text{cosine\_similarity}(Z^{(l)}, E^{(l)})$
3: **for** $o = 1$ to $O$ **do**
4:    $\mu_o \leftarrow \text{mean}(S^{(l)}[:, o])$
5:    $\sigma_o \leftarrow \text{std}(S^{(l)}[:, o])$
6:    $\text{threshold}_o \leftarrow \mu_o + \alpha \cdot \sigma_o$
7:    $S_{\text{filtered}}^{(l)}[:, o] \leftarrow (S^{(l)}[:, o] \geq \text{threshold}_o)$
8: **end for**
9: **Stage 2: High-scale Refinement**
10: $S^{(h)} \leftarrow \text{cosine\_similarity}(Z^{(h)}, E^{(h)})$
11: **for** $r = 1$ to $R$ **do**
12:    $n \leftarrow \text{parent\_patch\_index}(r)$ {Parent patch index}
13:    **for** $s = 1$ to $S$ **do**
14:      $o \leftarrow \text{parent\_text\_index}(s)$ {Parent text index}
15:      $S_{\text{masked}}^{(h)}(r, s) \leftarrow S^{(h)}(r, s) \cdot S_{\text{filtered}}^{(l)}(n, o)$
16:    **end for**
17: **end for**
18: **for** $s = 1$ to $S$ **do**
19:    $\mu_s \leftarrow \text{mean}(S_{\text{masked}}^{(h)}[:, s])$
20:    $\sigma_s \leftarrow \text{std}(S_{\text{masked}}^{(h)}[:, s])$
21:    $\text{threshold}_s \leftarrow \mu_s + \alpha \cdot \sigma_s$
22:    $S_{\text{filtered}}^{(h)}[:, s] \leftarrow (S_{\text{masked}}^{(h)}[:, s] \geq \text{threshold}_s)$
23: **end for**
24: **return** $S_{\text{filtered}}^{(l)}, S_{\text{filtered}}^{(h)}$

---

# E   Message Passing Details

HiVE-MIL introduces a newly designed hierarchical heterogeneous graph to jointly capture structural differences across scales and modalities. Previous approaches fail to explicitly encode hierarchical, coarse-to-fine structures across scales. To address this limitation, we apply relation-specific message passing operators to both intra-scale and hierarchical relationships.

The message passing process is as follows. During neighbor aggregation, information is collected from neighboring nodes connected via a specific relation type (i.e., edge type). These include *intra-scale relations*, such as valid patch-text connections at low or high scale, and *hierarchical relations*, such as connections between low-scale and high-scale patches, or between low-scale and high-scale text. The aggregated neighbor features are then concatenated with the node's own features and passed through a relation-specific linear transformation (i.e., weight matrix). For intra-scale relations, we adopt the GraphSAGE [21] operator via the HeteroConv module, whereas for hierarchical relations, we employ a Modality-Scale Attention (MSA) mechanism that accounts for both scale directionality and modality type. Finally, a non-linear activation function (ReLU) is applied to enhance representational capacity. By decoupling both aggregation and transformation parameters by relation type, our model effectively encodes heterogeneous and hierarchical information.

# F   Hierarchical Text Semantic Alignment Evaluation (Hit Ratio)

---

**Algorithm 2** Hierarchical Text Semantic Alignment Evaluation (Hit Ratio@2)

---

**Require:** Low-scale patch features $Z^{(l)} \in \mathbb{R}^{N \times D}$, high-scale patch features $Z^{(h)} \in \mathbb{R}^{N \times M \times D}$
   Low-scale text features $E^{(l)} \in \mathbb{R}^{O \times D}$, high-scale text features $E^{(h)} \in \mathbb{R}^{S \times D}$
   Hierarchical map $\mathcal{H} : \{1, \dots, O\}$ {Maps parent index to list of child indices}
**Ensure:** Hit Ratio@2 $\in [0, 1]$
1:   $hit \leftarrow 0$
2:   **for** $n = 1$ to $N$ **do**
3:      $z_n^{(l)} \leftarrow Z^{(l)}$
4:      $s^{(l)} \leftarrow \text{cosine\_similarity}(z_n^{(l)}, E^{(l)}) \in \mathbb{R}^O$
5:      $[o_1, o_2] \leftarrow \text{Top2Indices}(s^{(l)})$
6:      $C_n \leftarrow \mathcal{H}(o_1) \cup \mathcal{H}(o_2)$ {Candidate child indices}
7:      **for** $m = 1$ to $M$ **do**
8:         $z_{n,m}^{(h)} \leftarrow Z^{(h)}$
9:         $s^{(h)} \leftarrow \text{cosine\_similarity}(z_{n,m}^{(h)}, E^{(h)}) \in \mathbb{R}^S$
10:        $s^* \leftarrow \arg\max_s s^{(h)}$
11:        **if** $s^* \in C_n$ **then**
12:           $hit \leftarrow hit + 1$
13:           **break** {One hit is enough}
14:        **end if**
15:     **end for**
16:  **end for**
17:  **return** Hit Ratio@2 $= hit/N$

---

To assess whether the model preserves the intended hierarchical structure between coarse (low-scale) and fine-grained (high-scale) textual semantics, we introduce a quantitative evaluation metric termed *Hit Ratio@2*. While the main paper presents the high-level idea, we describe here the full procedure and rationale consistent with the implementation in Algorithm 2. The core objective is to evaluate whether the high-scale patch features selected by the model semantically align with the child-level text descriptions that are hierarchically linked to the most relevant low-scale (parent) prompts. Each low-scale ($5\times$) patch is associated with a parent-level textual description (e.g., *Diffuse Infiltration*) and each parent prompt is linked to a set of fine-grained child prompts (e.g., *Intracytoplasmic Mucin*, *Lack of Glandular Structures*) via a predefined hierarchical mapping $\mathcal{H} : \{1, \dots, O\}$.

**Step-by-step protocol.** For each low-scale patch embedding $z_n^{(l)} \in Z^{(l)}$, we first compute cosine similarities with all low-scale text features $E^{(l)}$, yielding a similarity vector $s^{(l)} \in \mathbb{R}^O$. We then

select the indices of the top-2 most similar parent prompts, denoted $o_1$ and $o_2$. The corresponding candidate child indices are obtained as $C_n = \mathcal{H}(o_1) \cup \mathcal{H}(o_2)$.

Next, for each high-scale patch $z_{n,m}^{(h)}$ within the spatial region of the $n$-th low-scale patch, we compute cosine similarities to all high-scale text features $E^{(h)}$, resulting in a score vector $s^{(h)} \in \mathbb{R}^S$. We identify the top-1 most similar child text index $s^* = \arg\max_s s^{(h)}$. A *hit* is recorded if $s^* \in C_n$ and we immediately terminate the further comparisons for the remaining high-scale patches in that region (i.e., only one hit is counted per low-scale patch). The final metric is computed as the ratio of low-scale patches with at least one correct child-level alignment among the top-2 parent candidates:

$$\text{Hit Ratio@2} = \frac{\text{\# of low-scale patches with a valid child match}}{N}$$

This procedure provides a controlled and interpretable evaluation of hierarchical alignment across visual-textual levels and validates whether the model's fine-grained decisions respect the structural guidance implied by the coarse-scale semantics.

## G   Spatial vs. Semantic Connectivity

Although HiVE-MIL does not explicitly encode spatial adjacency, it does capture structural context *indirectly* through a hierarchical and semantic design. At the $20\times$ scale, each high-resolution patch is linked to its corresponding $5\times$ parent patch via predefined absolute-coordinate mappings, enabling spatial alignment to be reflected hierarchically. At both $5\times$ and $20\times$ scales, semantically related patch–text pairs are selected using TGDF, allowing patches to form connections through shared textual descriptions. These semantic-based connections serve as an *intermediate* mechanism for modeling inter-patch relationships beyond physical proximity.

This design reflects the characteristics of pathological images, where spatially adjacent patches often differ in tissue structure, while semantically meaningful relationships can occur across distant regions. Therefore, we construct the intra-scale graph using semantic filtering, which allows connections between distant patches that share similar meanings based on textual descriptions. We argue that simple distance-based connections may overlook semantically important patterns.

## H   Implementation Details

We set the logit scaling factor $\gamma$ to 4.5871 (PLIP [25]), 4.6052 (QuiltNet [26]), and 4.0315 (CONCH [39]), using the values provided by each pre-trained VLM. We retain the top-2 patches at $5\times$ and the top-100 patches at $20\times$ scales to compute the final logits. Pooling- and MIL-based methods use only the image encoder, while VLM-based MIL methods leverage both image and text encoders. Most methods operate on single-scale $20\times$ patches, whereas DSMIL [33], ViLa-MIL [48], MSCPT [22], and HiVE-MIL **(Ours)** utilize multi-scale inputs from both $5\times$ and $20\times$ patches.

Our implementation is based on the official ViLa-MIL codebase [48] and all baselines as well as HiVE-MIL are implemented within this unified framework to ensure fair comparison. Our graph-based modules are implemented using PyTorch Geometric [13]. All experiments are conducted on Ubuntu 20.04.6 using a workstation equipped with two NVIDIA A100 GPUs (40 GB each); however, only one GPU is used for training each model. Complete package versions and dependencies are listed in the `requirements.txt` file available in our GitHub repository (linked in the Abstract).

# I  Baselines

We compare HiVE-MIL and other baseline models using image and text encoders from the following recent vision-language pathology foundation models.

- **PLIP [25]**: A vision-language pathology model pretrained on OpenPath, a large dataset of 208,414 image–text pairs curated from public platforms like medical Twitter, enabling strong zero-shot classification and case retrieval via image or language queries.

- **QuiltNet [26]**: A vision-language model pretrained on Quilt, a 1M-pair dataset curated from YouTube, medical Twitter, and academic sources, enabling strong zero-shot classification and cross-modal retrieval across diverse histopathology datasets.

- **CONCH [39]**: A recent state-of-the-art vision-language pathology foundation model, pretrained on 1.17 million image–caption pairs from diverse biomedical sources, enabling strong zero-shot and transferable performance across 14 benchmarks spanning classification, segmentation, captioning, and retrieval tasks.

We provide details for the baselines used for the few-shot WSI classification tasks.

- **ABMIL [27]**: An attention-based pooling framework that assigns adaptive weights to instances for more informative bag-level aggregation.

- **DSMIL [33]**: A dual-stream MIL model where one stream identifies a critical instance via max pooling, while the other aggregates instances by distance-weighted similarity.

- **CLAM [38]**: A weakly-supervised MIL framework that combines attention pooling with instance-level clustering to enhance interpretability and slide-level prediction.

- **TransMIL [47]**: A transformer-based MIL model that captures inter-instance correlations by modeling both spatial and morphological relationships among patches.

- **DTFD-MIL [54]**: A double-tier MIL framework that introduces pseudo-bags to improve training diversity and derives instance probabilities under an attention-based setting.

- **WiKG [34]**: A graph-based MIL approach that constructs WSIs as knowledge graphs with directed edges and updates features using knowledge-aware attention.

- **ViLa-MIL [48]**: A dual-scale VLM-MIL model generates LLM-based prompts and uses a prototype-guided image decoder and a context-guided text decoder.

- **MSCPT [22]**: A multi-scale VLM MIL model that integrates multi-scale visual inputs with LLM-generated prompts through graph-based reasoning and cross-scale aggregation.

- **FOCUS [19]**: A single-scale VLM-MIL model that integrates pathology foundation models with language priors for focused analysis of diagnostic regions.

# J  Baselines for Main Ablation Studies

This section introduces all baselines used in the ablation studies presented in the main paper.

### J.1  HTCL Variant Baselines

**No-Contrastive.**  In the No-contrastive variant, only the standard cross-entropy loss is used; no additional loss, such as the hierarchical text contrastive loss (HTCL), is applied.

**Share-Parent.**  In the Share-Parent variant, the anchor is defined as the mean embedding of all high-scale children sharing the same parent, i.e., $\bar{\mathbf{T}}_i^{(h)} = \frac{1}{|C(i)|} \sum_{s \in C(i)} \mathbf{T}_s^{(h)}$, where $C(i)$ denotes the set of high-scale children for parent node $i$. The similarity between the anchor and each low-scale (parent) embedding is computed as $\text{sim}_{i,j} = \cos(\bar{\mathbf{T}}_i^{(h)}, \mathbf{T}_j^{(l)})$. Positive pairs are defined as $\mathcal{P}_i = \{j \mid \text{Parent}(i) = j\}$, and negative pairs as $\mathcal{N}_i = \{j \mid \text{Parent}(i) \neq j\}$. This approach encourages the semantic alignment of all the children nodes under the same parent.

**Instance-Wise.** In the Instance-Wise variant, each high-scale embedding serves as the anchor. The similarity is calculated as $\text{sim}_{i,j} = \cos(\mathbf{T}_i^{(h)}, \mathbf{T}_j^{(l)})$, with positives and negatives defined by the parent relationship, i.e., $\mathcal{P}_i = \{j \mid \text{Parent}(i) = j\}$ and $\mathcal{N}_i = \{j \mid \text{Parent}(i) \neq j\}$. This strategy makes the embedding of each instance to be more similar to its own parent node.

## J.2 Module Components

**No TGDF.** Disables the model's ability to suppress irrelevant or weakly aligned patch-text pairs during graph construction. As a result, all patch and text nodes within each scale are densely connected in the intra-scale graph, irrespective of semantic relevance. This lack of filtering introduces spurious and noisy edges, which can degrade intra-scale alignment quality and propagate noise during message passing.

**No HTCL.** Removes explicit supervision that aligns parent and child text embeddings across scales. The model is then trained solely with standard cross-entropy loss, without constraints that enforce hierarchical consistency between coarse- and fine-scale textual semantics. Consequently, the embeddings of low- and high-scale text nodes may become semantically misaligned, thereby weakening hierarchical text semantic alignment.

**No HHG.** Eliminates all hierarchical edges connecting low- and high-scale nodes in both the visual and textual branches. The model operates with only intra-scale message passing, without leveraging the hierarchical structure. This prevents the flow of contextual information across scales and inhibits the modeling of coarse-to-fine semantic relationships between $5\times$ and $20\times$ features.

## J.3 Hierarchical Aggregator Baselines

**Scale-aware Attention (SAA).** To disentangle the contribution of the modality, we define scale-aware attention by removing the modality-specific transformation (i.e., using shared weights $W_q$, $W_k$, $W_v$ for all relations). The vectors are computed as $q_v = W_q(h_v + s_v)$, $k_u = W_k(h_u + s_u)$, $v_u = W_v(h_u + s_u)$, and the output is

$$h_v^{\text{scale}} = q_v + \sum_{u \in \mathcal{N}_r(v)} \beta_{vu} v_u, \quad \text{where} \quad \beta_{vu} = \text{softmax}\left(\frac{q_v^\top k_u}{\sqrt{d}}\right) \tag{10}$$

**Modality-aware Attention (MAA).** To isolate the effect of scale, we define modality-aware attention by removing the scale embedding (i.e., not adding $s_v$ or $s_u$). The vectors are computed as $q_v = W_q^{(r)} h_v$, $k_u = W_k^{(r)} h_u$, $v_u = W_v^{(r)} h_u$, and the output is

$$h_v^{\text{mod}} = q_v + \sum_{u \in \mathcal{N}_r(v)} \beta_{vu} v_u, \quad \text{where} \quad \beta_{vu} = \text{softmax}\left(\frac{q_v^\top k_u}{\sqrt{d}}\right) \tag{11}$$

**Attn.** For this Attn. variant, we apply standard attention [52] without any modifications, disregarding both scale and modality types.

**GraphSAGE.** For hierarchical-scale message passing, we adopt the standard GraphSAGE [21] aggregation without any additional attention mechanism or modifications.

## K    Generalization to Camelyon16

To evaluate generalizability beyond TCGA datasets (BRCA, NSCLC, RCC), we further test HiVE-MIL and baselines on the Camelyon16 dataset [5], using the same experimental settings as described in the main paper. HiVE-MIL achieves the highest performance, outperforming the second-best model, ViLa-MIL, by 1.25% in Macro F1.

Table 8: Performance comparison on the Camelyon16 dataset (CONCH, 16-shot).

| Camelyon16 [5] | | | |
|---|---|---|---|
| Model | ACC | AUC | Macro F1 |
| ABMIL [27] | 88.33 ±4.44 | 92.97 ±4.64 | 87.50 ±4.67 |
| DSMIL [33] | 90.33 ±2.77 | 94.95 ±1.82 | 89.92 ±2.85 |
| CLAM-SB [38] | 85.17 ±14.16 | 85.07 ±20.02 | 80.46 ±22.48 |
| DTFD-MIL (AFS) [38] | 91.17 ±4.25 | 93.35 ±3.36 | 91.92 ±4.43 |
| ViLa-MIL [48] | 92.33 ±5.04 | 96.37 ±2.28 | 91.97 ±5.22 |
| MSCPT [22] | 88.50 ±8.79 | 88.43 ±13.44 | 88.29 ±13.03 |
| FOCUS [19] | 90.83 ±4.44 | 91.62 ±6.52 | 90.23 ±4.84 |
| **HiVE-MIL** | **93.33 ±4.86** | **96.72 ±3.57** | **93.22 ±5.06** |

## L    Comparison with WSI Foundation Model

We conduct an additional experiment to compare against MADELEINE [29], a representative WSI foundation model (not MIL-based), as it is pre-trained on CONCH patch features to generate slide-level embeddings. This setup ensures a fair comparison with our method (CONCH + HiVE-MIL).

Table 9: HiVE-MIL vs. WSI Foundation Model (CONCH, 16-shot).

| Model | TCGA NSCLC | | | TCGA BRCA | | |
|---|---|---|---|---|---|---|
| | ACC | AUC | Macro F1 | ACC | AUC | Macro F1 |
| MADELEINE [29] (Linear Probing) | 83.00 ±4.10 | 90.30 ±3.50 | 83.00 ±4.10 | 80.40 ±6.70 | 88.40 ±6.40 | 80.30 ±6.70 |
| **HiVE-MIL** | **90.39 ±1.57** | **96.49 ±0.56** | **90.37 ±1.58** | **87.29 ±2.83** | **93.86 ±0.89** | **87.24 ±2.85** |

## M    Further Ablations

### M.1    Text Format

As detailed in Appendix A.3, the input text generated by the LLM follows a structured format consisting of a term and an explanation, e.g., *Dyskeratotic Cells* (term) and *Isolated eosinophilic tumor cells undergoing premature keratinization, appearing as dense, glassy bodies within cell clusters* (explanation). To evaluate the robustness of HiVE-MIL to different textual input formats, we conduct ablation studies using three variants: (i) term only, (ii) explanation only (denoted as *Exp.*), and (iii) term + explanation, which serves as the default format in our main experiments. As shown in Table 10, the term + explanation format achieves the highest performance on both the TCGA NSCLC and BRCA datasets. The term-only and explanation-only variants also perform competitively, with only marginal degradation. These results confirm that HiVE-MIL is robust to variations in textual input structure and suggest that combining concise terms with descriptive context yields more effective guidance.

Table 10: Text format ablation (CONCH, 16-shot).

| Text Format | TCGA NSCLC | | | TCGA BRCA | | |
|---|---|---|---|---|---|---|
| | ACC | AUC | Macro F1 | ACC | AUC | Macro F1 |
| *Term* | 89.55 ±4.27 | **96.92 ±0.70** | 89.50 ±4.34 | 86.64 ±2.01 | **94.22 ±1.12** | 86.57 ±2.00 |
| *Exp.* | 90.00 ±2.83 | 96.47 ±0.86 | 89.99 ±2.85 | 86.77 ±2.80 | 93.35 ±1.01 | 86.66 ±2.93 |
| **Term + Exp.** | **90.39 ±1.57** | 96.49 ±0.56 | **90.37 ±1.58** | **87.29 ±2.83** | 93.86 ±0.89 | **87.24 ±2.85** |

## M.2 Robustness to LLM Variants

To evaluate the robustness of HiVE-MIL to variations in textual input, we generate descriptions using four different LLMs: DeepSeek R1 [18], Grok 3 [53], Gemini 2.5 Pro [3], and GPT-4o [1]. The results in Table 11 demonstrate that HiVE-MIL performs robustly across all LLMs. Although GPT-4o yields the best overall results, the performance differences among the LLMs are relatively small, indicating that HiVE-MIL does not heavily depend on a specific LLM. Importantly, regardless of the LLM used to generate textual descriptions, HiVE-MIL consistently outperforms all baseline methods. We expect performance to improve further as pathology-specific LLMs become available in the future, since the generated texts will be more accurate and less biased.

Table 11: LLM variants ablation (CONCH, 16-shot).

| LLM | TCGA NSCLC | | | TCGA BRCA | | |
| --- | --- | --- | --- | --- | --- | --- |
| | ACC | AUC | Macro F1 | ACC | AUC | Macro F1 |
| *DeepSeek R1 [18]* | 89.36 ±3.88 | 95.77 ±1.68 | 89.33 ±3.92 | 85.60 ±4.22 | 93.08 ±2.43 | 85.49 ±4.36 |
| *Grok 3 [53]* | 90.19 ±2.33 | 96.33 ±1.01 | 90.18 ±2.33 | 85.80 ±3.27 | 93.62 ±1.80 | 85.71 ±3.32 |
| *Gemini 2.5 Pro [3]* | 90.00 ±2.52 | 96.47 ±0.99 | 89.98 ±2.55 | 86.65 ±2.10 | 93.08 ±1.79 | 86.61 ±2.10 |
| **GPT-4o [1]** | **90.39 ±1.57** | **96.49 ±0.56** | **90.37 ±1.58** | **87.29 ±2.83** | **93.86 ±0.89** | **87.24 ±2.85** |

## M.3 TGDF Component

To evaluate the contribution of individual components in the Text-Guided Dynamic Filtering (TGDF) module, we conduct ablation studies by selectively disabling its submodules. The first variant (*w/o Mask Prop. + Low Fil.*) removes both cross-scale mask propagation and threshold-based filtering at the low scale, retaining only normalization for the low-scale similarity matrix while preserving high-scale thresholding. The second variant (*w/o Mask Prop.*) disables only the cross-scale mask propagation, keeping threshold filtering active at both scales. The full TGDF configuration incorporates all components: cross-scale mask propagation, low-scale filtering, and high-scale filtering. As shown in Table 12, this complete TGDF module achieves the best performance across the TCGA NSCLC and BRCA datasets. These results confirm that both low-scale filtering and high-scale guidance via propagated masks are essential for effective multi-scale visual-textual alignment.

Table 12: TGDF component ablation (QuiltNet, 16-shot).

| | TCGA NSCLC | | | TCGA BRCA | | |
| --- | --- | --- | --- | --- | --- | --- |
| | ACC | AUC | Macro F1 | ACC | AUC | Macro F1 |
| *TGDF (w/o Mask Prop. + Low Fil.)* | 76.55 ±4.43 | 84.67 ±3.92 | 76.06 ±4.40 | 75.79 ±4.27 | 83.95 ±4.06 | 75.46 ±4.24 |
| *TGDF (w/o Mask Prop.)* | 77.63 ±7.00 | 85.51 ±6.04 | 77.57 ±7.04 | 76.63 ±3.42 | 84.11 ±2.30 | 76.45 ±3.47 |
| **TGDF (Ours)** | **79.23 ±2.70** | **87.34 ±4.08** | **79.09 ±2.75** | **77.08 ±3.90** | **84.31 ±4.22** | **76.80 ±4.15** |

## M.4 GNN vs. Alternative Interaction Models

HiVE-MIL is designed to model WSIs that inherently involve heterogeneous modalities (e.g., visual and textual features) and multiple scales (e.g., 5× and 20×). To reason over both intra-scale interactions and coarse-to-fine hierarchical relationships, the model must operate on complex and structured representations. GNNs offer a strong inductive bias for this purpose, as they are well suited to capturing such structured dependencies [4, 58]. Through localized message passing conditioned on edge types, GNNs enable HiVE-MIL to semantically align visual and textual nodes while preserving structural granularity across scales. In contrast, attention- or MLP-based approaches are limited in their capacity to effectively capture dependencies between nodes, and they struggle to incorporate prior structural knowledge, such as semantic links across modalities or hierarchical relationships across scales.

To verify the necessity of GNNs, we replace them with MLP or attention modules in both the intra-scale and hierarchical components and report the results in Table 13. The comparison confirms that GNNs play a crucial role in integrating intra-scale and hierarchical information, while simpler alternatives fall short in capturing these relationships effectively.

Table 13: GNN vs. alternative interaction models (PLIP, 16-shot).

| Module | | TCGA NSCLC | | | TCGA BRCA | | |
|---|---|---|---|---|---|---|---|
| Intra-scale | Hierarchical | ACC | AUC | Macro F1 | ACC | AUC | Macro F1 |
| MLP | MLP | 69.04 ±10.55 | 76.15 ±8.56 | 65.69 ±16.73 | 63.91 ±2.50 | 58.88 ±3.51 | 58.96 ±3.52 |
| Attention | Attention | 71.62 ±9.83 | 78.54 ±7.96 | 68.24 ±15.96 | 67.76 ±3.04 | 69.24 ±4.03 | 64.11 ±3.82 |
| Graph | MLP | 74.21 ±9.11 | 81.04 ±7.36 | 71.13 ±15.18 | 71.67 ±3.57 | 80.56 ±4.55 | 71.03 ±4.11 |
| Graph | Attention | 76.99 ±3.23 | 84.50 ±3.46 | 76.89 ±3.24 | 73.11 ±3.83 | 81.41 ±2.24 | 72.96 ±3.55 |
| **Graph** | **HGNN** | **80.13 ±4.73** | **87.28 ±2.76** | **80.08 ±4.73** | **75.21 ±3.51** | **83.19 ±4.72** | **74.99 ±3.67** |

## M.5 Single-Scale vs. Multi-Scale Interaction

To assess the contribution of scale-level interaction, we evaluate three configurations: (a) using only low-scale features, (b) using only high-scale features, and (c) combining both through multiscale integration. Table 14 highlights the consistent superiority of the multi-scale configuration over both single-scale variants on the TCGA NSCLC and BRCA datasets. Low-scale features primarily encode coarse structural context, while high-scale features capture fine-grained morphological detail. Limiting the model to a single scale restricts its ability to fully exploit the semantic richness of WSI data. The superior performance achieved by the multi-scale setting validates our design choice to model hierarchical relationships and highlights the importance of integrating complementary information across scales in visual–textual alignment.

Table 14: Single vs. multi-scale interaction (CONCH, 16-shot).

| Row | Instance | Logit | TCGA NSCLC | | | TCGA BRCA | | |
|---|---|---|---|---|---|---|---|---|
| | Low High | Low High | ACC | AUC | Macro F1 | ACC | AUC | Macro F1 |
| (a) | ✓ – ✗ | ✓ – ✗ | 84.99 ±2.10 | 89.23 ±1.53 | 84.92 ±2.17 | 79.17 ±3.49 | 85.45 ±1.13 | 78.97 ±3.56 |
| (b) | ✗ – ✓ | ✗ – ✓ | 86.17 ±2.22 | 91.33 ±1.05 | 86.15 ±2.22 | 83.17 ±2.90 | 87.36 ±1.03 | 83.05 ±2.97 |
| (c) | ✓ – ✓ | ✓ – ✓ | **90.39 ±1.57** | **96.49 ±0.56** | **90.37 ±1.58** | **87.29 ±2.83** | **93.86 ±0.89** | **87.24 ±2.85** |

# N  Hyperparameter Sensitivity

## N.1  TGDF Hyperparameter

The TGDF module uses a hyperparameter $\alpha$ to compute the threshold that determines the sensitivity of patch–text alignment filtering at both low and high scales. Specifically, $\alpha$ influences the number of retained entries in the similarity matrices $S_{\text{filtered}}^{(l)}$ and $S_{\text{filtered}}^{(h)}$, thereby affecting which visual–textual pairs are preserved for downstream processing. Table 15 show how overall performance varies with different TGDF filtering thresholds. To clarify, $\alpha = 0$ does not imply "no TGDF"; rather, it sets the filtering threshold as $\mu + \alpha \cdot \sigma$, meaning the threshold equals $\mu$ when $\alpha = 0$. To compute the proportion of filtered patch–text pairs, we calculate the filtered-pair ratio per WSI, average it across WSIs, and report the mean over five runs.

When TGDF is not applied at all, performance is lowest due to aligning all image–text pairs, which results in incorrect image–text pairings. As $\alpha$ increases, more patch–text pairs are filtered out at both scales, leading to improved performance, with $\alpha = 0.5$ yielding the best result, indicating that TGDF effectively removes semantically irrelevant image–text pairs. However, when $\alpha = 1$, filtering becomes too aggressive, discarding many meaningful pairs and reducing performance.

Table 15: TGDF threshold $\alpha$ hyperparameter sensitivity (QuiltNet, 16-shot).

| $\alpha$ | TCGA NSCLC | | TCGA BRCA | |
|---|---|---|---|---|
| | Filtered Pairs (5×/20×) | Macro F1 | Filtered Pairs (5×/20×) | Macro F1 |
| No TGDF | 0% / 0% | 75.13 ± 4.21 | 0% / 0% | 74.24 ± 5.27 |
| -1 | 11.58% / 17.21% | 77.02 ± 3.25 | 12.46% / 16.85% | 75.12 ± 4.10 |
| -0.5 | 28.25% / 30.02% | 78.52 ± 3.67 | 30.11% / 31.05% | 76.17 ± 4.68 |
| 0 | 46.34% / 51.23% | 79.06 ± 4.40 | 48.53% / 52.45% | 76.00 ± 3.47 |
| **0.5** | **67.02% / 70.65%** | **79.09 ± 2.75** | **67.92% / 71.80%** | **76.80 ± 4.15** |
| 1 | 85.21% / 86.03% | 78.55 ± 4.52 | 84.16% / 85.43% | 76.09 ± 3.41 |

### N.2 HTCL Hyperparameter

The total loss is defined as the sum of the standard cross-entropy loss (CE) and the hierarchical text contrastive loss (HTCL), i.e., $\mathcal{L}_{\text{total}} = \mathcal{L}_{\text{CE}} + \lambda \mathcal{L}_{\text{HTCL}}$, where $\lambda$ controls the influence of HTCL. To assess the sensitivity of the model performance to this hyperparameter, we perform an ablation study by varying $\lambda \in \{0, 0.1, 0.5, 1\}$. Incorporating HTCL ($\lambda > 0$) consistently improves performance over the baseline ($\lambda = 0$), confirming the benefit of hierarchical text supervision (Table 16). All non-zero values yield comparable results, with $\lambda = 0.5$ achieving the best overall performance. These results demonstrate that HTCL is robust to hyperparameter selection and effective at guiding multi-scale visual-textual alignment.

Table 16: HTCL $\lambda$ hyperparameter sensitivity (PLIP, 16-shot).

| $\lambda$ | TCGA NSCLC | | | TCGA BRCA | | |
|---|---|---|---|---|---|---|
| | ACC | AUC | Macro F1 | ACC | AUC | Macro F1 |
| 0 | 78.14 ±3.55 | 86.02 ±3.69 | 78.11 ±3.54 | 73.96 ±4.42 | 80.50 ±6.02 | 73.81 ±4.46 |
| 0.1 | 79.30 ±4.00 | 86.19 ±3.30 | 79.26 ±3.98 | 74.50 ±3.76 | 82.08 ±6.51 | 74.22 ±3.91 |
| **0.5** | **80.13 ±4.73** | **87.28 ±2.76** | **80.08 ±4.73** | **75.21 ±3.51** | **83.19 ±4.72** | **74.99 ±3.67** |
| 1 | 79.46 ±3.78 | 86.62 ±3.13 | 78.37 ±3.78 | 75.16 ±3.61 | 82.94 ±3.34 | 74.94 ±3.68 |

### N.3 Top-$k$ Logit Hyperparameter

To further assess the robustness of HiVE-MIL with respect to the top-$k$ hyperparameters used for logit computation, we conduct a sensitivity analysis by varying the number of selected patches at both scales. Specifically, we vary $K \in \{2, 5, 10\}$ at the low scale and $N \in \{50, 100, 200\}$ at the high scale. The results are shown in Figure 8, using a 3D bar plot to depict performance in all combinations of $(K, N)$. The variation in performance across the nine configurations is minimal, indicating that HiVE-MIL is robust and largely insensitive to the specific choice of $K$ and $N$ within the tested ranges.

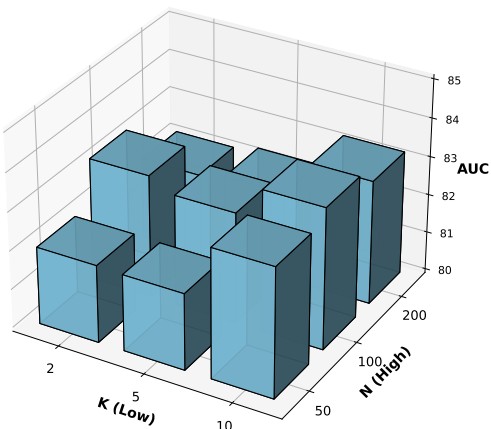

Figure 8: AUC sensitivity to top-$k$ logit selection at low ($K$) and high ($N$) scales (TCGA BRCA, PLIP, 16-shot). HiVE-MIL shows stable performance across all $(K, N)$ combinations.

# O Computational Efficiency & Scalability

We evaluate HiVE-MIL along with comparison models (2024 onwards) in terms of FLOPs, inference time, and maximum GPU memory usage. The computation time is measured using an NVIDIA A100 GPU. To evaluate runtime overhead in a realistic setting, we first use a WSI from TCGA BRCA containing 1,880 patches at $5\times$ and 28,249 patches at $20\times$ (Left). For scalability, we also report results on the largest WSI in the dataset, which contains 2,633 patches at $5\times$ and 40,777 patches at $20\times$ (Right).

Table 17: Computational efficiency and scalability.

| Model | WSI 1 ($5\times$: 1,880, $20\times$: 28,249 patches) | | | WSI 2 ($5\times$: 2,633, $20\times$: 40,777 patches) | | |
|---|---|---|---|---|---|---|
| | FLOPs (G) | Inference Time (s) | Max GPU Memory Usage (MB) | FLOPs (G) | Inference Time (s) | Max GPU Memory Usage (MB) |
| *WiKG [34]* | 891.82 | 0.0775 | 4630.56 | 1810.44 | 0.1539 | 8617.82 |
| *ViLa-MIL [48]* | 87.65 | 0.0242 | 528.20 | 116.00 | 0.0199 | 679.41 |
| *MSCPT [22]* | 860.21 | 0.1189 | 2074.89 | 894.17 | 0.1368 | 2695.92 |
| *FOCUS [19]* | 39.52 | 1.4500 | 353.21 | 46.20 | 1.9484 | 429.51 |
| **HiVE-MIL** | 624.74 | 0.2037 | 4738.36 | 738.26 | 0.2583 | 6456.60 |

**Optimizations planned.** Working with WSIs is undeniably challenging due to their gigapixel size. To make our method more practical for large-scale deployment, we will further develop a parallel version of HiVE-MIL that can better handle the computational load. This would help reduce processing time and make the model more efficient for deployment in clinical settings.

# P Potential Societal Impact

**Positive Impacts.** HiVE-MIL enables data-efficient, few-shot classification of WSIs by modeling hierarchical dependencies and intra-scale multimodal alignments through a unified hierarchical heterogeneous graph. This approach has the potential to improve diagnostic support in resource-limited settings, where annotated datasets and expert pathologists are often unavailable. By aligning visual patches with descriptive prompts on multiple scales, the model improves interpretability and can help clinicians make more informed decisions. Beyond pathology, hierarchical design should also benefit other domains that involve limited data and structured semantic inputs.

**Negative Impacts.** The model relies on LLM-generated prompts to construct hierarchical text descriptions, which may introduce factual errors or biases, particularly in the absence of expert oversight. In clinical settings, excessive reliance on automatically generated outputs without independent review and validation can lead to biased or misleading results. This underscores the importance of expert participation in ensuring both the accuracy and reliability of the model's explanations.

