# OpenReview forum: "Few-Shot Learning from Gigapixel Images via Hierarchical Vision-Language Alignment and Modeling"
_NeurIPS.cc/2025/Conference — NeurIPS 2025 poster_

### Official Review · Reviewer_BfPp · 2025-06-27

**Clarity:** 3
**Significance:** 2
**Originality:** 2
**Rating:** 4
**Confidence:** 4

**Summary:**

HiVE-MIL proposes a few-shot learning framework for whole-slide image classification that integrates multi-scale image patches with LLM-generated textual descriptions. By constructing a hierarchical heterogeneous graph linking visual and textual features across scales, and applying a lightweight graph neural network with modality-scale attention, the method achieves strong performance on TCGA benchmarks while maintaining parameter efficiency and offering interpretable outputs.

**Questions:**

1. Are the disease-specific textual descriptions generated by the LLM verified by pathologists for accuracy?
2. The experiments are limited to TCGA datasets. Could the authors validate the generalizability of their method on external datasets such as CAMELYON or PANDA?

**Ethical Concerns:**

["NO or VERY MINOR ethics concerns only"]

**Final Justification:**

Thanks for the valuable feedback, it well address my concerns, so I will raise my score to 4.

**Limitations:**

Please see weaknesses and questions.

**Paper Formatting Concerns:**

No formatting concerns.

**Quality:**

3

**Strengths And Weaknesses:**

### Strengths

1. This paper is well written and easy to follow.
2. The design of TGDF combined with hierarchical graphs enhances the model's interpretability. Additionally, the use of a lightweight graph head and frozen VLM encoders makes the model parameter-efficient and stable under few-shot settings.
3. The experimental comparison is comprehensive, and the ablation studies are also very thorough.

### Weaknesses

1. In line 57, the authors claim that *"current models do not explicitly model the interactions between visual and textual features on the same scale."* However, this claim is not entirely accurate. Existing works such as ViLa-MIL and MSCPT already incorporate cross-modal feature interaction at the same scale. For example, ViLa-MIL (Figure 2) includes a context-guided text decoder and slide-text similarity computation; MSCPT (Figure 2) introduces image-text similarity-based graph prompt tuning. This weakens the paper’s stated motivation and contribution.
2. In Equation 7, the index notation is confusing, mixing the letters *i* and *c*. Moreover, the explanation of text features in the methods section only accounts for low-scale feature count *O* and high-scale feature count *K* or *S*, but neglects the class number *C* (e.g. Equation 4). This sudden appearance of *C* in Equation 7 may confuse readers.
3. The modeling of image features only considers hierarchical relationships between low and high scales, without incorporating spatial adjacency within the same scale. This may hinder the model from capturing visually recognizable patterns such as gland–stroma–inflammatory cell arrangements.
4. The fixed threshold α in TGDF is heuristic and may not generalize across datasets or backbones. An inappropriate α could result in the omission of rare but critical tumor regions, reducing diagnostic reliability.

---

> ### Author Rebuttal · Authors · 2025-07-31
>
> Thank you for your positive feedback and for recognizing that our paper is well written and easy to follow. We also appreciate your acknowledgment of the model’s interpretability and the comprehensiveness of our experiments. We provide the following responses to address your concerns.
>
> | **W1. Clarification regarding motivation and contribution**|
> |--- |
> We apologize for the confusion. What we intended to convey is that while current models such as ViLa-MIL and MSCPT do perform cross-modal feature interaction at the same scale, the alignment between visual and textual features remains *inadequate*. This point is reflected in line 7 (Abstract) and line 57, which state: *“Inadequate alignment between modalities on the same scale.”* Existing models do not *fully* explore the interactions between different modalities in constructing task-specific knowledge. In contrast, our method addresses this limitation through a heterogeneous graph that explicitly models semantic connections between visual and textual nodes at the same scale (line 74).
>
> We acknowledge that the sentence referenced by the reviewer may have unintentionally implied that existing models lack any form of cross-modal modeling, which was not our intention. We will revise the wording in the final version to clearly reflect our intended message.
>
> | **W2. Clarification regarding equation**|
> |--- |
> We thank the reviewer for pointing out the confusion in Equation 7 regarding the index notation and the insufficient explanation of class-wise text prompt grouping.
>
> We agree that the use of indices in Equation 7 was potentially confusing, especially due to the mixing of the image index $i$ and class index $c$. In the revised version, we will clarify the notation by introducing a class-specific text index set $\mathcal{I}_c$, where each $i \in \mathcal{I}_c$ corresponds to a text prompt associated with class $c$. This allows us to express the class logit computation more clearly. In the equation below, the $ \mathrm{TopKAvg}_k$ function computes the average of the top $k$ similarity scores from the $i$-th column (.,i) of the image-text similarity matrix. This step aggregates the signal from the most relevant images for each text prompt before the final summation.
>
>
> `logit_c^{(s)} = (γ / |I_c|) * Σ_{i∈I_c} TopKAvg_k([X^{(s)} (T^{(s)})ᵗ]_{·,i})`
>
>
> This avoids index overloading and makes the operation easier to follow.
>
>
>
> | **W3. Not incorporating spatial adjacency**|
> |---  |
> Thank you for your valuable feedback. Although HiVE-MIL does not explicitly encode spatial adjacency, it does capture structural context *indirectly* through a hierarchical and semantic design. At the 20x scale, each high-resolution patch is linked to its corresponding 5x parent patch via predefined absolute-coordinate mappings, enabling spatial alignment to be reflected hierarchically. At both 5x and 20x scales, semantically related patch–text pairs are selected using TGDF, allowing patches to form connections through shared textual descriptions. These semantic-based connections serve as an *intermediate* mechanism for modeling inter-patch relationships beyond physical proximity.
>
> This design reflects the characteristics of pathological images, where spatially adjacent patches often differ in tissue structure, while semantically meaningful relationships can occur across distant regions. Therefore, we construct the intra-scale graph using semantic filtering, which allows connections between distant patches that share similar meanings based on textual descriptions. We argue that simple distance-based connections may overlook semantially important patterns. Nevertheless, we acknowledge the reviewer’s suggestion to explicitly incorporate intra-scale spatial adjacency as a valuable extension and we plan to explore ways to integrate spatial and semantic relationships in parallel in future work.
>
> | **W4. Heuristic α may omit rare tumors**|
> |---  |
> We appreciate the reviewer’s concern that the fixed threshold α in the TGDF module could potentially miss rare but critical tumor regions, thereby reducing diagnostic reliability. However, we would like to clarify that the purpose of α is not to filter or suppress tumor regions, but rather to prevent irrelevant image patches from being connected to unrelated textual prompts, such as background patches paired with disease-specific text, which could confuse the model during learning. In other words, TGDF filters out semantically meaningless image–text pairs and retains only the meaningful ones, which are then used to form edges in the heterogeneous graph, thereby encouraging the model to focus on learning coherent and semantically rich visual–language associations.
>
> Furthermore, as shown in the sensitivity analysis presented in Supplementary Material H.2, we evaluated a range of α values (0, 0.1, 0.5, 1.0) and observed that the model’s performance remains relatively stable across settings. The default value α \= 0.5 yields the most consistent results overall. That said, we fully acknowledge the reviewer’s point that a fixed threshold may not be optimal across all scenarios. As such, we plan to explore adaptive, learnable filtering mechanisms in future work to further improve generalization across datasets or backbones.
>
> | **Q1. Pathologists verification**|
> |---  |
> Thank you for asking this important question. While the descriptions generated by the LLM have not yet been verified by pathologists, we address this limitation through a series of alternative validation strategies designed to assess their class-discriminative relevance and semantic reliability.
>
> (1) We evaluate the *faithfulness* of the generated descriptions using an "LLM-as-a-Judge" \[1\], where LLMs are prompted to infer class labels based solely on the generated descriptions. We employ two  evaluators: GPT-4o and GPT o3, to assess descriptions generated by several LLMs: Deep Seek R1, Grok 3, Gemini 2.5 Pro, and GPT-4o. Although not perfect, the number of correct predictions is still consistently high, suggesting that the generated text is generally reliable while capturing class-discriminative signals.
>
> | |  | **Evaluator: GPT-4o** |  | **Evaluator: GPT o3** |  |
> | :---: | :---: | :---: | :---: | :---: | :---: |
> | **Generator**  |**\#Total Descriptions (\#5x/\#20x)**  | **TCGA NSCLC** | **TCGA BRCA** | **TCGA NSCLC** | **TCGA BRCA** |
> | DeepSeek R1 | 32 (8/24) | 28 (8/20) | 28 (8/20) | 31 (8/23) | 28 (8/20) |
> | Grok 3 | 32 (8/24) | 28 (7/21) | 28 (8/20) | 32 (8/24) | 21 (8/23) |
> | Gemini 2.5 Pro | 32 (8/24) | 30 (8/22) | 29 (7/22) | 30 (7/23) | 32 (8/24) |
> | GPT-4o | 32 (8/24) | 30 (8/22) | 30 (7/23) | 31 (8/23) | 32 (8/24) |
> | Total | 128 (32/96) | 116 (31/85) | 114 (29/85) | 124 (31/93) | 113 (32/89) |
>
> \[1\] Gu et al. A Survey on LLM-as-a-Judge, arXiv 2025\.
>
> (2) The table below reports intra- and inter-category text similarity scores, where *intra* denotes similarities among descriptions of the same morphology or concept, and *inter* denotes similarities across different ones. Similarity is computed separately at each scale and jointly across both scales using the CONCH text encoder and cosine similarity. These results highlight that the descriptions encode highly discriminative and consistent semantics for each morphological or conceptual group, with robustness across scales and LLMs.
>
>
>
> || **TCGA NSCLC** |  |  | **TCGA BRCA** |  |  |
> | :---: | :---: | :---: | :---: | :---: | :---: | :---: |
> | **LLM**  | **5x (intra/inter)** | **20x (intra/inter)** | **5x,20x (intra/inter)** | **5x (intra/inter)** | **20x (intra/inter)** | **5x, 20x (intra/inter)** |
> | DeepSeek R1 | 1.000 (0.157) | 0.515 (0.164) | 0.456 (0.158) | 1.000 (0.165) | 0.438 (0.141) | 0.412 (0.138) |
> | Grok 3 | 1.000 (0.123) | 0.511 (0.170) | 0.434 (0.160) | 1.000 (0.186) | 0.490 (0.214) | 0.415 (0.212) |
> | Gemini 2.5 Pro | 1.000 (0.111) | 0.558 (0.165) | 0.478 (0.150) | 1.000 (0.206) | 0.491 (0.127) | 0.426 (0.142) |
> | GPT-4o | 1.000 (0.066) | 0.516 (0.114) | 0.454 (0.109) | 1.000 (0.163) | 0.477 (0.108) | 0.413 (0.125) |
>
> Furthermore, we validated the performance of HiVE-MIL using descriptions generated by different LLMs, as detailed in Supplementary Material H.5.
>
> | **Q2. Generalization beyond TCGA datasets**|
> |---  |
> To evaluate generalizability beyond TCGA datasets (BRCA, NSCLC, RCC), we further tested HiVE-MIL and baselines on the **Camelyon16 dataset**, using the same experimental settings as described in the main paper. HiVE-MIL achieves the highest performance, outperforming the second-best model, ViLa-MIL, by 1.25% in Macro F1 (CONCH, 16-shot).
>
>
>
> | **Model**           | **ACC**           | **AUC**           | **Macro F1**      |
> |---------------------|-------------------|-------------------|-------------------|
> | ABMIL               | 88.33 ± 4.44      | 92.97 ± 4.64      | 87.50 ± 4.67      |
> | DSMIL               | 90.33 ± 2.77      | 94.95 ± 1.82      | 89.92 ± 2.85      |
> | CLAM-SB             | 85.17 ± 14.16     | 85.07 ± 20.02     | 80.46 ± 22.48     |
> | DTFD-MIL (AFS)      | 91.17 ± 4.25      | 93.35 ± 3.36      | 91.92 ± 4.43      |
> | ViLa-MIL            | 92.33 ± 5.04      | 96.37 ± 2.28      | 91.97 ± 5.22      |
> | MSCPT               | 88.50 ± 8.79      | 88.43 ± 13.44     | 88.29 ± 13.03     |
> | FOCUS               | 90.83 ± 4.44      | 91.62 ± 6.52      | 90.23 ± 4.84      |
> | **HiVE-MIL**        | **93.33 ± 4.86**  | **96.72 ± 3.57**  | **93.22 ± 5.06**  |
>
>
> Again, thank you for your thoughtful and constructive comments.

---

> > ### Comment · Reviewer_BfPp · 2025-08-05
> > **Thanks**
> >
> > Thanks for the valuable feedback, it well address my concerns, so I will raise my score to 4.

---

### Official Review · Reviewer_FSfv · 2025-06-28

**Clarity:** 2
**Significance:** 3
**Originality:** 3
**Rating:** 4
**Confidence:** 4

**Summary:**

This work introduces HiVE‑MIL, a novel framework for few‑shot, weakly supervised whole‑slide image classification that builds a hierarchical vision‑language graph to jointly model coarse‑to‑fine tissue structures and align visual/textual modalities; it incorporates a text‑guided dynamic filtering module and a hierarchical contrastive loss to refine patch–text pairings and enforce cross‑scale semantic consistency, and on TCGA breast, lung, and kidney cancer datasets under 4‑, 8‑, and 16‑shot settings, HiVE‑MIL outperforms both traditional and recent VLM‑based MIL methods by up to 4.1 % in macro F1 (16‑shot) .

**Questions:**

1.The paper introduces a relation‑specific GraphSAGE operator but does not detail its internal steps. Could you provide a concise illustration or pseudocode showing how neighbor aggregation, relation‑specific transformations, and activation functions are composed within this operator?
2.Your Text‑Guided Dynamic Filtering (TGDF) and Hierarchical Text‑Contrastive Loss (HTCL) rely on textual features generated by a vision‑language model. Could you elaborate on the exact VLM used, the prompt‐engineering process, and how sensitive your results are to different prompt templates or VLM backbones?
3.GNN necessity: The model relies on a hierarchical graph neural network to model inter‑ and intra‑scale interactions—could you clarify why a GNN is the most appropriate mechanism here, and whether alternative interaction schemes (e.g., attention‑based fusion or message passing via simpler MLPs) were considered?
4.You set the TGDF filtering threshold α=0.5 by default—can you share how this value was chosen, what proportion of patch–text pairs it typically filters out, and how sensitive overall performance is to variations in α?

**Ethical Concerns:**

["NO or VERY MINOR ethics concerns only"]

**Final Justification:**

My concerns have been resolved. The paper is technically acceptable.

**Limitations:**

While proposing a GNN‑based hierarchical graph, the paper lacks discussion on why graph neural networks are necessary versus simpler fusion mechanisms. The authors should acknowledge this choice’s assumptions and consider evaluating alternative interaction models.

**Paper Formatting Concerns:**

No issues

**Quality:**

3

**Strengths And Weaknesses:**

Strengths
1.Thoughtful integration of modalities: The paper is the first to integrate a hierarchical heterogeneous graph structure with vision‑language models (VLMs), enabling coarse‑to‑fine modeling of tissue architecture in WSIs.
2.Sound design: The combination of Text‑Guided Dynamic Filtering (TGDF) and Hierarchical Text‑Contrastive Loss (HTCL) effectively improves patch–text alignment and enforces cross‑scale semantic consistency.
3.Comprehensive evaluation: Experiments on TCGA breast, lung, and kidney cancer datasets under 4‑, 8‑, and 16‑shot settings show consistent performance gains, with up to a 4.1 % macro F1 improvement in the 16‑shot scenario.
Weaknesses
1.Lack of theoretical justification for using GNNs: The paper does not sufficiently argue why a graph neural network is the necessary choice to model the proposed relationships, nor whether alternative interaction mechanisms might be more efficient.
2.Insufficient hyperparameter analysis: Key settings such as the TGDF threshold (α = 0.5) are not justified; it is unclear what proportion of patch–text pairs this value filters and how sensitive performance is to this choice.
3.Dependence on text quality: The effectiveness of TGDF and HTCL hinges on the quality of the generated textual prompts; any biases or inaccuracies in VLM‑generated text could degrade performance.

---

> ### Author Rebuttal · Authors · 2025-07-31
>
> Thank you for your positive feedback and for recognizing the thoughtful integration of modalities, sound design, and comprehensive evaluation in our work. We provide the following responses to address your concerns.
>
> | **W1/Q3/L1. Justification for using GNNs**|
> |--- |
> We appreciate the reviewer’s thoughtful question. HiVE-MIL is designed to model WSIs that inherently involve heterogeneous modalities (e.g., visual and textual features) and multiple scales (e.g., 5x and 20x). To reason over both intra-scale interactions and coarse-to-fine hierarchical relationships, the model must operate on complex and structured representations. GNNs offer a strong inductive bias for this purpose, as they are well suited to capturing such structured dependencies \[1,2\]. Through localized message passing conditioned on edge types, GNNs enable HiVE-MIL to semantically align visual and textual nodes while preserving structural granularity across scales.
>
> In contrast, attention- or MLP-based approaches are limited in their capacity to effectively capture dependencies between nodes, and they struggle to incorporate prior structural knowledge, such as semantic links across modalities or hierarchical relationships across scales.
>
> \[1\] Battaglia et al. Relational inductive biases, deep learning, and graph networks, arXiv 2018\.
> \[2\] Zhou et al. Graph Neural Networks: A Review of Methods and Applications, AI Open 2021\.
>
> **GNN vs. alternative interaction models.** To verify the necessity of GNNs, we replace them with MLP or attention modules in both the intra-scale and hierarchical components and report the results below (PLIP, 16-shot). The comparison confirms that GNNs play a crucial role in integrating intra-scale and hierarchical information, while simpler alternatives fall short in capturing these relationships effectively.
>
> |  |  |  |  |  |  |  |  |
> | :---: | :---: | :---: | :---: | :---: | :---: | :---: | :---: |
> | **Module** |  | **TCGA NSCLC** |  |  | **TCGA BRCA** |  |  |
> | **Intra-scale** | **Hierarchical** | **ACC** | **AUC** | **Macro F1** | **ACC** | **AUC** | **Macro F1** |
> | MLP | MLP | 69.04 ± 10.55  | 76.15 ± 8.56 | 65.69 ± 16.73 | 63.91 ± 2.50 | 58.88 ± 3.51 | 58.96 ± 3.52 |
> | Attention | Attention | 71.62 ± 9.83 | 78.54 ± 7.96 | 68.24 ± 15.96 | 67.76 ± 3.04 | 69.24 ± 4.03 | 64.11 ± 3.82 |
> | Graph | MLP | 74.21 ± 9.11 | 81.04 ± 7.36 | 71.13 ± 15.18 | 71.67 ± 3.57 | 80.56 ± 4.55 | 71.03 ± 4.11 |
> | Graph | Attention | 76.99 ± 3.23 | 84.50 ± 3.46 | 76.89 ± 3.24 | 73.11 ± 3.83 | 81.41 ± 2.24 | 72.96 ± 3.55  |
> | **Graph** | **HGNN** | **80.13 ± 4.73**  | **87.28 ± 2.76** | **80.08 ± 4.73** | **75.21 ± 3.51** | **83.19 ± 4.72** | **74.99 ± 3.67** |
>
> We will add these rationales and experimental results to the final version of the paper.
>
> | **Q1. Details on the message passing**|
> |--- |
> We apologize for not clearly detailing the steps. This study introduces a newly designed heterogeneous hierarchical graph to jointly capture structural differences across modalities and scales. While prior approaches may distinguish between modalities, they often fail to explicitly encode hierarchical, coarse-to-fine structures across scales. To address this limitation, we apply relation-specific message passing operators to both intra-scale and hierarchical relationships.
>
> The message passing process is as follows. During neighbor aggregation, information is collected from neighboring nodes connected via a specific relation type (i.e., edge type). These include *intra-scale relations*, such as valid patch-text connections at low or high scale, and *hierarchical relations*, such as connections between low-scale and high-scale patches, or between low-scale and high-scale text. The aggregated neighbor features are then concatenated with the node’s own features and passed through a relation-specific linear transformation (i.e., weight matrix). For intra-scale relations, we adopt the GraphSAGE operator via the HeteroConv module, whereas for hierarchical relations, we employ a Modality-Scale Attention (MSA) mechanism that accounts for both scale directionality and modality type. Finally, a non-linear activation function (ReLU) is applied to enhance representational capacity. By decoupling both aggregation and transformation parameters by relation type, our model effectively encodes heterogeneous and hierarchical information. The corresponding pseudocode will be included in the final version.
>
> | **W2/Q4. TGDF filtering threshold**|
> |--- |
> The results below (QuiltNet, 16-shot) shows how overall performance varies with different TGDF filtering thresholds. To clarify, α \= 0 does not imply "no TGDF"; rather, it sets the filtering threshold as μₒ \+ α·σₒ (Section 3.2, Main Paper), meaning the threshold \= μₒ when α \= 0\. To compute the proportion of filtered patch-text pairs, we first calculate the ratio of filtered pairs for each WSI and then take the mean across WSIs. The reported value is the average of these means over five independent runs.
>
> When TGDF is not applied at all, performance is lowest due to aligning all image-text pairs, which would result in incorrect image-text pairings. As α increases, more patch-text pairs are filtered out at both scales, leading to improved performance, with α \= 0.5 yielding the best result, indicating that TGDF effectively removes semantically irrelevant image–text pairs. However, when α \= 1, the filtering becomes too aggressive, discarding many meaningful pairs and reducing performance.
>
>
>
> || **TCGA NSCLC** |    | **TCGA BRCA** |  |
> | :---: | :---: | :---: | :---: | :---: |
>  |  **α**|   **Filtered Pairs (5x/20x)** |   **Macro F1** | **Filtered Pairs (5x/20x)** |  **Macro F1** |
> | No TGDF  | 0% / 0% | 75.13 ± 4.21 | 0% / 0% | 74.24 ± 5.27 |
>  | \-1 | 11.58% / 17.21% | 77.02 ± 3.25 | 12.46%/16.85% | 75.12 ± 4.10 |
> |  \-0.5 | 28.25% / 30.02% | 78.52 ± 3.67 | 30.11% / 31.05% | 76.17 ± 4.68 |
> |   0 | 46.34% / 51.23% | 79.06 ± 4.40 | 48.53% / 52.45% | 76.00 ± 3.47 |
> |   0.5 | 67.02% / 70.65% | 79.09 ± 2.75 | 67.92% / 71.80% | 76.80 ± 4.15 |
> |  1 | 85.21% / 86.03% | 78.55 ± 4.52 | 84.16% / 85.43% | 76.09 ± 3.41 |
>
> | **Q2. Details on text generation**|
> |--- |
> Thank you for the insightful question. We would like to first clarify that the textual features used in TGDF and HTCL are generated by the LLM, not VLM. Specifically, we use GPT-4o as the primary model (note: we also experimented with several other LLMs, as shown below) to generate text descriptions using the prompt detailed in Section 3.1 of the main paper. For your convenience, we also include the prompt here:
>
> > The task is to summarize the morphological features of the `{dataset_name}` dataset for the `{class_name_1}`, `{class_name_2}`, … classes. For each class, list **O** representative morphological features observed at **5×** magnification, followed by **K** finer sub-features observed at **20×** magnification for each. Each description should include the morphological term along with an explanation of its defining visual features.
>
> The examples of the descriptions generated by the LLM are provided in Supplementary Material A.3.
>
> | **W3/Q2. Sensitivity to prompt template and LLM backbone**|
> |---|
> To assess sensitivity to textual inputs, we conducted two sets of robustness analyses:
>
> (1) **Prompt Template Sensitivity** (Supplementary Material H.4): We evaluated three formats: Term-only, Explanation-only (Exp.), and Term \+ Explanation (Term \+ Exp.). The Term \+ Explanation format consistently achieves the best performance, suggesting that combining concise terms with descriptive context yields more effective guidance (CONCH, 16-shot).
>
>
>
> |  | **TCGA NSCLC** |  |  | **TCGA BRCA** |  |  |
> | :---: | :---: | :---: | :---: | :---: | :---: | :---: |
> | **Prompt Template** | **ACC** | **AUC** | **Macro F1** | **ACC** | **AUC** | **Macro F1** |
> | Term | 89.55 ±4.27 | **96.92 ±0.70** | 89.50 ±4.34 | 86.64 ±2.01 | **94.22 ±1.12** | 86.57 ±2.00 |
> | Exp. | 90.00 ±2.83 | 96.47 ±0.86 | 89.99 ±2.85 | 86.77 ±2.80 | 93.35 ±1.01 | 86.66 ±2.93 |
> | **Term \+ Exp.** | **90.39 ±1.57** | 96.49 ±0.56 | **90.37 ±1.58** | **87.29 ±2.83** | 93.86 ±0.89 | **87.24 ±2.85** |
>
> (2) **LLM Backbone Sensitivity** (Supplementary Material H.5) : We generated descriptions using four different LLMs: DeepSeek R1, Grok 3, Gemini 2.5 Pro, and GPT-4o. GPT-4o achieves the best overall performance. The results are consistent across different LLMs, demonstrating that HiVE-MIL is robust to the choice of LLM backbone (CONCH, 16-shot).
>
>
>
> | | **TCGA NSCLC** |  |  | **TCGA BRCA** |  |  |
> | :---: | :---: | :---: | :---: | :---: | :---: | :---: |
> |  **LLM** | **ACC** | **AUC** | **Macro F1** | **ACC** | **AUC** | **Macro F1** |
> | DeepSeek R1 | 89.36 ±3.88 | 95.77 ±1.68  | 89.33 ±3.92 | 85.60 ±4.22 | 93.08 ±2.43 | 85.49 ±4.36 |
> | Grok 3 | 90.19 ±2.33 | 96.33 ±1.01 | 90.18 ±2.33 | 85.80 ±3.27 | 93.62 ±1.80 | 85.71 ±3.32 |
> | Gemini 2.5 Pro | 90.00 ±2.52 | 96.47 ±0.99 | 89.98 ±2.55 | 86.65 ±2.10 | 93.08 ±1.79 | 86.61 ±2.10 |
> | **GPT-4o** | **90.39 ±1.57** | **96.49 ±0.56** | **90.37 ±1.58** | **87.29 ±2.83** | **93.86 ±0.89** | **87.24 ±2.85** |
>
> Again, we thank you for your thoughtful and constructive comments.

---

> > ### Author Response · Authors · 2025-08-05
> >
> > Thank you again for your valuable feedback. We would like to confirm whether our previous response has fully addressed your concerns and answered your questions. If there is anything still unclear or any concerns we have not yet addressed, please let us know and we will be happy to provide further clarification.

---

> > > ### Author Response · Authors · 2025-08-08
> > > **Follow-Up on Response to Reviewer FSfv**
> > >
> > > Dear Reviewer FSfv,
> > >
> > > As the discussion period is ending soon, we would like to follow up to confirm whether our response has addressed your concerns. If there are any remaining questions or issues, we would greatly appreciate the opportunity to discuss them further. Thank you again for your time and consideration.

---

> > > > ### Author Response · Authors · 2025-08-09
> > > > **Follow-up Before Discussion Period Ends (8 Hours Remaining)**
> > > >
> > > > Dear Reviewer FSfv,
> > > >
> > > > As the discussion period will end in about 8 hours, we would like to follow up to confirm whether our response has addressed your concerns. If there are any remaining questions or issues, we would greatly appreciate the opportunity to discuss them further. Thank you again for your time and consideration.

---

> > ### Comment · Reviewer_FSfv · 2025-08-09
> >
> > I appreciate the additional analyses and experiments provided in the rebuttal, which offer some clarifications. I look forward to seeing them further refined and integrated into the camera-ready paper.

---

> > > ### Author Response · Authors · 2025-08-09
> > >
> > > Thank you for your feedback and for acknowledging our additional analyses and experiments. We’re glad the rebuttal has addressed your concerns and we will further refine and integrate them in the camera-ready paper.

---

### Official Review · Reviewer_S9yU · 2025-06-29

**Clarity:** 3
**Significance:** 3
**Originality:** 3
**Rating:** 5
**Confidence:** 3

**Summary:**

The paper proposes HiVE-MIL, a hierarchical vision-language framework for few-shot weakly supervised classification of whole slide images (WSIs) in computational pathology. Existing methods often fail to model the hierarchical relationships within visual and textual modalities across different magnifications and lack effective alignment between modalities at the same scale. To overcome these limitations, HiVE-MIL constructs a hierarchical heterogeneous graph that incorporates parent–child links between low- and high-magnification nodes and connects visual and textual nodes within the same scale. The results demonstrate the value of jointly modeling hierarchical structure and multimodal alignment for efficient and scalable learning from limited pathology data.

**Questions:**

- What is the impact of the vision-language backbone choice? The paper evaluates HiVE-MIL with multiple pretrained VLMs, but the architectural changes are tightly coupled with their embedding quality. How sensitive is the proposed framework to the backbone choice?
- Could the authors elaborate on computational efficiency and scalability? The proposed hierarchical graph structure and attention mechanisms introduce additional complexity compared to standard MIL. Can the authors provide more insight into the runtime and memory overhead, especially for larger WSIs? Are there any bottlenecks or optimizations planned for deployment in clinical settings?

**Ethical Concerns:**

["NO or VERY MINOR ethics concerns only"]

**Final Justification:**

The paper is solid and my concerns are solved by the authors' rebuttal. My final rating is 5: Accept.

**Limitations:**

Yes.

**Paper Formatting Concerns:**

No major formatting issues in this paper.

**Quality:**

4

**Strengths And Weaknesses:**

**Strengths:**
- The experimental evaluation is comprehensive, covering three TCGA datasets and multiple vision-language models. Ablation studies and other analysis further support the robustness and effectiveness of the proposed method.
- The proposed HiVE-MIL framework integrates well-established components with novel adaptations.
- The motivation is clear, and the methodology is logically developed.

**Weaknesses:**
- While the empirical results are strong, the reliance on frozen vision-language backbones may limit its adaptability.

In summary, the paper has high technical quality, clear contributions, and practical relevance, though future work may need to address limitations in adaptivity and complexity.

---

> ### Author Rebuttal · Authors · 2025-07-31
>
> We thank you for the positive feedback and for recognizing the high technical quality, clear contributions, and practical relevance of our work. We provide the following responses to address your concerns.
>
> | **W1. Reliance on frozen vision-language backbones**|
> |---|
> We appreciate the comment and agree that freezing the VLM backbones may limit their adaptability. Also, please note that, in few-shot settings, where only a small amount of training data is available and due to the risk of overfitting, it is common to use train-time adaptation methods such as CoOp \[1\], which adapt pre-trained (frozen) VLMs to downstream tasks by updating only a small number of parameters. Because methods like CoOp are effective only for single images (i.e., patch-level), we transfer this knowledge to the WSI level and leverage pathology-specific prior knowledge from the pre-trained VLMs by integrating CoOp with our proposed HiVE-MIL framework, which effectively models the hierarchical and contextual information present in WSIs.
>
> \[1\] Zhou et al. Learning to Prompt for Vision-Language Models, IJCV 2022\.
>
> | **Q1. On sensitivity to vision-language backbone choice**|
> |---|
> HiVE-MIL is designed to be modular and independent of any specific vision-language backbone. Its core components, including the hierarchical heterogeneous graph, TGDF, and HTCL, operate separately from the embedding source and consistently enhance representation quality. To assess sensitivity, we evaluated HiVE-MIL using three pathology-specific VLMs trained on progressively larger pre-training datasets: PLIP (208K pairs), QuiltNet (1M pairs), and CONCH (1.17M pairs). Across all of the backbones and datasets, HiVE-MIL consistently outperforms baseline methods. Even when using PLIP, which is pre-trained on the smallest dataset, HiVE-MIL achieves a 3.26% improvement in macro F1 over the second-best method on the TCGA NSCLC dataset (Table 1, main paper). These results demonstrate that HiVE-MIL is robust to variations in VLM quality. We also expect its performance to further improve as stronger pathology-specific VLMs become available.
>
> | **Q2. Computational efficiency and scalability**|
> |---|
> We have evaluated HiVE-MIL along with comparison models (2024 onwards) in terms of FLOPs, inference time, and maximum GPU memory usage. The computation time was measured using an NVIDIA A100 GPU. To evaluate runtime overhead in a realistic setting, we first used a WSI from TCGA BRCA containing 1,880 patches at 5x and 28,249 patches at 20x (Left). For scalability, we also report results on the largest WSI in the dataset, which contains 2,633 patches at 5x and 40,777 patches at 20x (Right).
>
>
>
> | | **# 5x: 1,880, #20x: 28,249** |  |  | **5x: 2,633, #20x: 40,777** |  |  |
> | :---: | :---: | :---: | :---: | :---: | :---: | :---: |
> | **Model** | **FLOPs (G)** | **Inference Time (s)** | **Maximum GPU Memory Usage (MB)** | **FLOPs (G)** | **Inference Time (s)** | **Maximum GPU Memory Usage (MB)** |
> | WiKG | 891.82 | 0.0775 | 4630.56 | 1810.44 | 0.1539 | 8617.82 |
> | ViLa-MIL | 87.65 | 0.0242 | 528.20 | 116 | 0.0199 | 679.41 |
> | MSCPT | 860.21 | 0.1189 | 2074.89 | 894.17 | 0.1368 | 2695.92 |
> | FOCUS | 39.52 | 1.4500 | 353.21 | 46.20 | 1.9484 | 429.51 |
> | HiVE-MIL | 624.74 | 0.2037 | 4738.36 | 738.26 | 0.2583 | 6456.6 |
>
> **Optimizations planned.** Working with WSIs is undeniably challenging due to their gigapixel size. To make our method more practical for large-scale deployment, we will further develop a parallel version of HiVE-MIL that can better handle the computational load. This would help reduce processing time and make the model more efficient for deployment in clinical settings.
>
> Again, we thank you for your thoughtful and constructive comments.

---

### Official Review · Reviewer_6Jif · 2025-07-04

**Clarity:** 3
**Significance:** 3
**Originality:** 3
**Rating:** 4
**Confidence:** 4

**Summary:**

This paper proposes HiVE-MIL, a hierarchical vision-language framework that constructs a unified graph with (1) parent-child links between coarse (5x) and fine (20x) visual/textual nodes to capture hierarchical relationships, and (2) heterogeneous intra-scale edges linking visual and textual nodes at the same scale. To enhance semantic consistency, HiVE-MIL incorporates a two-stage, text-guided dynamic filtering mechanism to remove weakly correlated patch-text pairs and introduces a hierarchical contrastive loss to align textual semantics across scales. Extensive experiments on TCGA breast, lung, and kidney cancer datasets demonstrate its effectiveness.

**Questions:**

Besides the concerns mentioned in the weaknesses section, there is another issue:
The description of coarse and fine morphological features is crucial in the proposed framework, and these features are generated by the LLM. Is there any validation or guarantee regarding the accuracy of the content generated by the LLM?

**Ethical Concerns:**

["NO or VERY MINOR ethics concerns only"]

**Final Justification:**

Given the original quality of the paper and the response experiments, this paper could be accepted.

**Limitations:**

yes

**Quality:**

3

**Strengths And Weaknesses:**

Strengths:
1) the authors did take the hierarchy of features (5X, 20X) into consideration, which is one of the most important things in analyzing pathological images.
2) there is a carefully designed graph-based module that could connect visual and textual nodes on the same scale
3) the models they compared with are selected from the top-tier journal and trained with large amount of data. The experiments could clearly demonstrate the superior performance of the proposed method.

Major Weaknesses:
1) the author only mentioned 5X and 20X. In real-world scenario, there are 10X and 40X. Is there any way of handling these?
2) there is some methods that could handle giga pixel without MIL-related method, like  Giga-Path. I would like to see some comparison experiment on those methods.

Minor Problems:
1) font size of Figure 2 is too small.

---

> ### Author Rebuttal · Authors · 2025-07-31
>
> We thank you for acknowledging the importance of our hierarchical design, the careful graph-based visual-textual modeling, and the strong experimental comparisons. We provide the following responses to address your concerns.
>
> | **W1. Other scales (10x and 40x)**|
> |---|
> We chose the (5x, 20x) combination because it is the **most commonly used setting** in multi-scale MIL research. This setup has been widely adopted in prior work, such as MSCPT \[1\]. Thus, we followed it for consistency and fair comparison. Moreover, our framework is flexible and can be easily adapted to other scale pairs, such as (10x, 40x), using the same hierarchical design.
>
> For the combination of using all of the four scales (5x, 10x, 20x, 40x), the proposed method can support this configuration, although adding more scales naturally increases model complexity, inference time, and memory usage, as is also the case for other multi-scale MIL models. However, HiVE-MIL mitigates this overhead through TGDF, which filters out semantically meaningless image–text pairs at each scale and retains only the meaningful ones for constructing heterogeneous intra-scale edges. TGDF also operates in a *top-down* manner, meaning that when a low-scale patch such as 5x is removed, all its corresponding higher-scale patches (10x, 20x, 40x) are automatically discarded, thereby avoiding unnecessary computation. The parent–child mapping between scales is determined by the square of the scale ratio. For example, a single 5x patch corresponds to 4 patches at 10x $( (10/5)^2 = 4 )$ and 16 patches at 20x $( (20/5)^2 = 16 )$. Additional text prompts can also be generated for each scale, although this requires creating separate scale-specific descriptions.
>
> \[1\] Han et al. MSCPT: Few-shot Whole Slide Image Classification with Multi-scale and Context-focused Prompt Tuning, IEEE TMI 2025\.
>
> | **W2. Comparison with non MIL-related method**|
> |---|
> Following the comment, we have conducted an additional experiment to make a comparison with MADELEINE \[2\] as the representative of the non-MIL related method instead of GigaPath for the following reasons:
>
> (1) GigaPath’s slide encoder requires 1536-dim inputs from its own patch encoder, which is incompatible with the 512-dim features (PLIP, QuiltNet, CONCH) used in our work.
>
> (2) MADELEINE is pre-trained on CONCH patch features to create a slide embedding, making the comparison with (CONCH + HiVE-MIL) fair.
>
> (3) MADELEINE has been shown to outperform GigaPath in its original paper, making it a stronger baseline.
>
> The experimental results are as follows (CONCH, 16-shot)**:**
>
> | | **TCGA NSCLC** |  |  | **TCGA BRCA** |  |  |
> | :---: | :---: | :---: | :---: | :---: | :---: | :---: |
> | **Model**  | **ACC** | **AUC** | **Macro F1** | **ACC** | **AUC** | **Macro F1** |
> | MADELEINE (Linear Probing) | 83.00 ± 4.10  | 90.30 ± 3.50 | 83.00 ± 4.10 | 80.40 ± 6.70 | 88.40 ± 6.40 | 80.30 ± 6.70 |
> | **HiVE-MIL** | **90.39 ± 1.57** | **96.49 ± 0.56** | **90.37 ± 1.58** | **87.29 ± 2.83** | **93.86 ± 0.89** | **87.24 ± 2.85** |
>
> Overall, our method significantly outperforms MADELEINE (Linear Probing) on both datasets. We will add these additional experimental results to our final version of the paper.
>
> \[2\] Jaume et al. Multistain Pretraining for Slide Representation Learning in Pathology, ECCV 2024\.
>
> | **Minor. Font size of Figure 2**|
> |-|
> Thank you for pointing this out. We will update the font size in Figure 2 to improve readability in the final version.
>
> | **Q1. Validation of LLM-generated descriptions**|
> |---|
> Thank you for asking this important question. While the descriptions generated by the LLM have not yet been verified by pathologists, we address this limitation through a series of alternative validation strategies designed to assess their class-discriminative relevance (1) and semantic reliability (2).
>
> (1) We evaluate the *faithfulness* of the generated descriptions using an "LLM-as-a-Judge" \[3\], where LLMs are prompted to infer class labels based solely on the generated descriptions. We employ two evaluators: GPT-4o and GPT o3, to assess descriptions generated by several LLMs: Deep Seek R1, Grok 3, Gemini 2.5 Pro, and GPT-4o. Although not perfect, the number of correct predictions is still consistently high, suggesting that the generated text is generally reliable while capturing class-discriminative signals.
>
>
>
> | |  | **Evaluator: GPT-4o** |  | **Evaluator: GPT o3** |  |
> | :---: | :---: | :---: | :---: | :---: | :---: |
> | **Generator**  |**\#Total Descriptions (\#5x/\#20x)**  | **TCGA NSCLC** | **TCGA BRCA** | **TCGA NSCLC** | **TCGA BRCA** |
> | DeepSeek R1 | 32 (8/24) | 28 (8/20) | 28 (8/20) | 31 (8/23) | 28 (8/20) |
> | Grok 3 | 32 (8/24) | 28 (7/21) | 28 (8/20) | 32 (8/24) | 21 (8/23) |
> | Gemini 2.5 Pro | 32 (8/24) | 30 (8/22) | 29 (7/22) | 30 (7/23) | 32 (8/24) |
> | GPT-4o | 32 (8/24) | 30 (8/22) | 30 (7/23) | 31 (8/23) | 32 (8/24) |
> | Total | 128 (32/96) | 116 (31/85) | 114 (29/85) | 124 (31/93) | 113 (32/89) |
>
> \[3\] Gu et al. A Survey on LLM-as-a-Judge, arXiv 2025\.
>
> (2) The table below reports intra- and inter-category text similarity scores, where *intra* denotes similarities among descriptions of the same morphology or concept, and *inter* denotes similarities across different ones. Similarity is computed separately at each scale and jointly across both scales using the CONCH text encoder and cosine similarity. These results highlight that the descriptions encode highly discriminative and consistent semantics for each morphological or conceptual group, with robustness across scales and LLMs.
>
>
>
> | | **TCGA NSCLC** |  |  | **TCGA BRCA** |  |  |
> | :---: | :---: | :---: | :---: | :---: | :---: | :---: |
> | **LLM**  | **5x (intra/inter)** | **20x (intra/inter)** | **5x,20x (intra/inter)** | **5x (intra/inter)** | **20x (intra/inter)** | **5x,20x (intra/inter)** |
> | DeepSeek R1 | 1.000 (0.157) | 0.515 (0.164) | 0.456 (0.158) | 1.000 (0.165) | 0.438 (0.141) | 0.412 (0.138) |
> | Grok 3 | 1.000 (0.123) | 0.511 (0.170) | 0.434 (0.160) | 1.000 (0.186) | 0.490 (0.214) | 0.415 (0.212) |
> | Gemini 2.5 Pro | 1.000 (0.111) | 0.558 (0.165) | 0.478 (0.150) | 1.000 (0.206) | 0.491 (0.127) | 0.426 (0.142) |
> | GPT-4o | 1.000 (0.066) | 0.516 (0.114) | 0.454 (0.109) | 1.000 (0.163) | 0.477 (0.108) | 0.413 (0.125) |
>
> Furthermore, we validated the performance of HiVE-MIL using descriptions generated by different LLMs, as detailed in Supplementary Material H.5.
>
> Again, we thank you for your thoughtful and constructive comments.

---

> > ### Author Response · Authors · 2025-08-05
> >
> > Thank you again for your valuable feedback. We would like to confirm whether our previous response has fully addressed your concerns and answered your questions. If there is anything still unclear or any concerns we have not yet addressed, please let us know and we will be happy to provide further clarification.

---

> > > ### Author Response · Authors · 2025-08-08
> > > **Follow-Up on Response to Reviewer 6Jif**
> > >
> > > Dear Reviewer 6Jif,
> > >
> > > As the discussion period is ending soon, we would like to follow up to confirm whether our response has addressed your concerns. If there are any remaining questions or issues, we would greatly appreciate the opportunity to discuss them further. Thank you again for your time and consideration.

---

> > > > ### Author Response · Authors · 2025-08-09
> > > > **Follow-up Before Discussion Period Ends (8 Hours Remaining)**
> > > >
> > > > Dear Reviewer 6Jif,
> > > >
> > > > As the discussion period will end in about 8 hours, we would like to follow up to confirm whether our response has addressed your concerns. If there are any remaining questions or issues, we would greatly appreciate the opportunity to discuss them further. Thank you again for your time and consideration.

---

### Note · Authors · 2025-08-12

We thank all reviewers for their constructive feedback. **Following the rebuttal, all reviewers expressed positive assessments and support for acceptance**. Below we summarize the key strengths and main additions/clarifications from the rebuttal.

**Key strengths acknowledged by reviewers**

* **High technical quality and relevance (S9yU):** High technical quality with a logically developed method, clear contributions and motivation, and strong practical relevance to pathology.

* **Thoughtful integration of modalities (FSfv, 6Jif):** First to integrate a hierarchical heterogeneous graph with VLMs for coarse-to-fine (hierarchical) WSI modeling, aligning visual/textual nodes within each scale.

* **Sound design (FSfv):** The combination of TGDF and HTCL effectively improves patch–text alignment and enforces cross‑scale semantic consistency.

* **Comprehensive evaluation (6Jif, S9yU, FSfv, BfPp):** Evaluated on three TCGA datasets (breast, lung, kidney) with multiple VLM backbones (PLIP, QuiltNet, CONCH) in few-shot (4-, 8-, 16-shot) settings.

* **Interpretability (BfPp):** The design of TGDF combined with hierarchical graphs enhances the model’s interpretability.

* **Clarity of presentation (BfPp):** The paper is well written and easy to follow.


**Key rebuttal additions & clarifications**

* **Validation of LLM-generated content (6Jif, BfPp):** Assessed *faithfulness* via “LLM-as-a-Judge” for class-discriminative relevance and measured intra-/inter-category similarity for semantic reliability.

* **Comparative validation (6Jif, BfPp):** Added MADELEINE as a strong non-MIL baseline; validated on Camelyon16 for generalization.

* **Computational efficiency and scalability (S9yU):** Reported runtime, FLOPs, and memory usage.

* **GNN design justification (FSfv):** Provided rationale for using GNNs; ablation studies replacing them with MLP or attention modules confirmed their necessity.

* **Robustness analyses (FSfv, BfPp):** Included hyperparameter analysis of TGDF α, prompt template sensitivity, and LLM backbone sensitivity.
* **Clarifications (BfPp):** Clarified the motivation over ViLa-MIL and MSCPT; fixed equation/notation issues; explained why same-scale spatial adjacency was not incorporated.


We thank the reviewers again for their constructive feedback and positive assessments. The rebuttal has effectively addressed their concerns, and the additional analyses, experiments, and clarifications will be incorporated into the camera-ready version.

---

### Decision · Program_Chairs · 2025-09-17

**Decision:**

Accept (poster)

**Comment:**

The paper addresses a current problem in Digital histopathology in modeling interactions across different scales when using VLMs. After rebuttal reviewers raised their scores and are all positive towards acceptance. The authors should try to incorporate their rebuttal clarifications to the final paper